# Demixed principal component analysis of neural population data

**Dmitry Kobak[1†], Wieland Brendel[1,2,3†], Christos Constantinidis[4], Claudia E Feierstein[1], Adam Kepecs[5], Zachary F Mainen[1], Xue-Lian Qi[4], Ranulfo Romo[6,7], Naoshige Uchida[8], Christian K Machens[1*]**

[1]Champalimaud Neuroscience Program, Champalimaud Centre for the Unknown, Lisbon, Portugal; [2]École Normale Supérieure, Paris, France; [3]Centre for Integrative Neuroscience, University of Tübingen, Tübingen, Germany; [4]Wake Forest University School of Medicine, Winston-Salem, United States; [5]Cold Spring Harbor Laboratory, Cold Spring Harbor, United States; [6]Instituto de Fisiología Celular-Neurociencias, Universidad Nacional Autónoma de México, Mexico City, Mexico; [7]El Colegio Nacional, Mexico City, Mexico; [8]Harvard University, Cambridge, United States

**Abstract** Neurons in higher cortical areas, such as the prefrontal cortex, are often tuned to a variety of sensory and motor variables, and are therefore said to display mixed selectivity. This complexity of single neuron responses can obscure what information these areas represent and how it is represented. Here we demonstrate the advantages of a new dimensionality reduction technique, demixed principal component analysis (dPCA), that decomposes population activity into a few components. In addition to systematically capturing the majority of the variance of the data, dPCA also exposes the dependence of the neural representation on task parameters such as stimuli, decisions, or rewards. To illustrate our method we reanalyze population data from four datasets comprising different species, different cortical areas and different experimental tasks. In each case, dPCA provides a concise way of visualizing the data that summarizes the task-dependent features of the population response in a single figure.

**\*For correspondence:** christian. machens@neuro.fchampalimaud. org

[†]These authors contributed equally to this work

## Introduction

In many state of the art experiments, a subject, such as a rat or a monkey, performs a behavioral task while the activity of tens to hundreds of neurons in the animal's brain is monitored using electro-physiological or imaging techniques. The common goal of these studies is to relate the external task parameters, such as stimuli, rewards, or the animal's actions, to the internal neural activity, and to then draw conclusions about brain function. This approach has typically relied on the analysis of single neuron recordings. However, as soon as hundreds of neurons are taken into account, the complexity of the recorded data poses a fundamental challenge in itself. This problem has been particularly severe in higher-order areas such as the prefrontal cortex, where neural responses display a baffling heterogeneity, even if animals are carrying out rather simple tasks (*Brody et al., 2003*; *Machens et al., 2010*; *Hernández et al., 2010*; *Mante et al., 2013*; *Rigotti et al., 2013*).

Traditionally, this heterogeneity has often been neglected. In neurophysiological studies, it is common practice to pre-select cells based on particular criteria, such as responsiveness to the same stimulus, and to then average the firing rates of the pre-selected cells. This practice eliminates much of the richness of single-cell activities, similar to imaging techniques with low spatial resolution, such as MEG, EEG, or fMRI. While population averages can identify some of the information that higher-order areas process, they ignore much of the fine structure of the single cell responses (*Wohrer et al., 2013*). Indeed, most neurons in higher cortical areas will typically encode several

**eLife digest** Many neuroscience experiments today involve using electrodes to record from the brain of an animal, such as a mouse or a monkey, while the animal performs a task. The goal of such experiments is to understand how a particular brain region works. However, modern experimental techniques allow the activity of hundreds of neurons to be recorded simultaneously. Analysing such large amounts of data then becomes a challenge in itself.

This is particularly true for brain regions such as the prefrontal cortex that are involved in the cognitive processes that allow an animal to acquire knowledge. Individual neurons in the prefrontal cortex encode many different types of information relevant to a given task. Imagine, for example, that an animal has to select one of two objects to obtain a reward. The same group of prefrontal cortex neurons will encode the object presented to the animal, the animal's decision and its confidence in that decision. This simultaneous representation of different elements of a task is called a 'mixed' representation, and is difficult to analyse.

Kobak, Brendel et al. have now developed a data analysis tool that can 'demix' neural activity. The tool breaks down the activity of a population of neurons into its individual components. Each of these relates to only a single aspect of the task and is thus easier to interpret. Information about stimuli, for example, is distinguished from information about the animal's confidence levels.

Kobak, Brendel et al. used the demixing tool to reanalyse existing datasets recorded from several different animals, tasks and brain regions. In each case, the tool provided a complete, concise and transparent summary of the data. The next steps will be to apply the analysis tool to new datasets to see how well it performs in practice. At a technical level, the tool could also be extended in a number of different directions to enable it to deal with more complicated experimental designs in future.

task parameters simultaneously, and therefore display what has been termed *mixed selectivity* (*Rigotti et al., 2013*; *Pagan and Rust, 2014*; *Park et al., 2014*; *Raposo et al., 2014*).

Instead of looking at single neurons and selecting from or averaging over a population of neurons, neural population recordings can be analyzed using dimensionality reduction methods (for a review, see *Cunningham and Yu, 2014*). In recent years, several such methods have been developed that are specifically targeted to electrophysiological data, working on the level of single spikes (*Pfau et al., 2013*), accommodating different time scales of latent variables (*Yu et al., 2009*), or accounting for the dynamics of the population response (*Buesing et al., 2012a*; *2012b*; *Churchland et al., 2012*). However, these approaches reduce the dimensionality of the data without taking task parameters, i.e., sensory and motor variables controlled or monitored by the experimenter, into account. Consequently, mixed selectivity remains in the data even after the dimensionality reduction step.

The problem can be addressed by dimensionality reduction methods that are informed by the task parameters (*Machens et al., 2010*; *Machens, 2010*; *Brendel et al., 2011*; *Mante et al., 2013*; *Raposo et al., 2014*). We have previously introduced a dimensionality reduction technique, *demixed principal component analysis (dPCA)* (*Brendel et al., 2011*), that emphasizes two goals. It aims to find a decomposition of the data into latent components that (a) are easily interpretable with respect to the experimentally controlled and monitored task parameters; and (b) preserve the original data as much as possible, ensuring that no valuable information is thrown away. Here we present a radically modified version of this method, and illustrate that it works well on a wide variety of experimental data. The new version of the method has the same objectives as the older version (*Brendel et al., 2011*), but is more principled, more flexible, and has an analytical solution, meaning that it does not suffer from any numerical optimization problems. Furthermore, the new mathematical formulation highlights similarities to and differences from related well-known methods such as principal component analysis (PCA) and linear discriminant analysis (LDA).

The dPCA code is available at http://github.com/machenslab/dPCA for Matlab and Python.

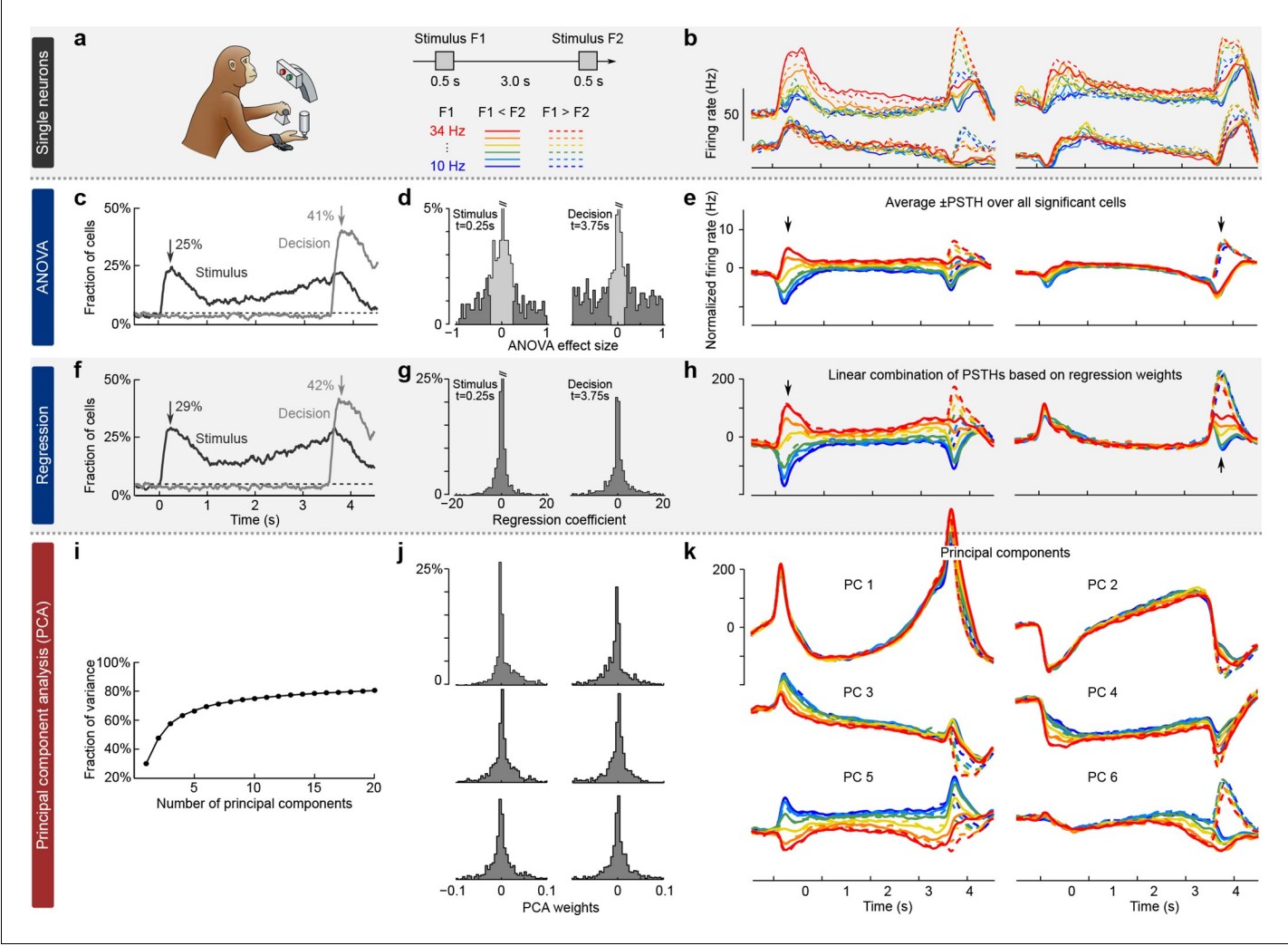

**Figure 1.** Existing approaches to population analysis, illustrated with recordings from monkey PFC during a somatosensory working memory task (*Romo et al., 1999*). (a) Cartoon of the paradigm, adapted from *Romo and Salinas (2003)*. Legend shows 12 experimental conditions. (b) Average per-condition firing rates (PSTHs) for four exemplary neurons out of $N = 832$. Colors refer to stimulus frequencies F1 and line styles (dashed/solid) refer to decision, see legend in (a). (c) Fraction of cells, significantly ($p<0.05$, two-way ANOVA) tuned to stimulus and decision at each time point. (d) Left: Distribution of stimulus tuning effect sizes across neural population at F1 period (black arrow in c). Significantly tuned neurons are shown in dark gray. Right: Same for decision at F2 period (gray arrow in c). (e) The average of zero-centered PSTHs over all significantly tuned neurons (for neurons with negative effect size, the sign of PSTHs was flipped). Arrows mark time-points that were used to select the significant cells. (f) Fraction of cells, significantly ($p<0.05$, linear regression) tuned to stimulus and decision at each time point. (g) Distribution of regression coefficients of neural firing rates to stimulus (during F1 period) and decision (during F2 period). (h) Stimulus and decision components produced by the method of *Mante et al. (2013)*. Briefly, neural PSTHs are weighted by the regression coefficients. (i) Fraction of variance captured by the first 20 principal components. (j) Distributions of weights used to produce the first six principal components (weights are elements of the eigenvectors of the $N \times N$ covariance matrix). (k) First six principal components (projections of the full data onto the eigenvector directions).

## Results

### Existing approaches

We illustrate the classical approaches to analyzing neural activity data from higher-order areas in *Figure 1*. To be specific, we consider recordings from the prefrontal cortex (PFC) of monkeys performing a somatosensory working memory task (*Romo et al., 1999*; *Brody et al., 2003*). In this task, monkeys were required to discriminate two vibratory stimuli presented to the fingertip. The stimuli F1 and F2 were separated by a 3 s delay, and the monkeys had to report which stimulus had a higher frequency by pressing one of the two available buttons (*Figure 1a*).

When we focus on the neural representation of the stimulus F1 and the decision, we have to take 12 experimental conditions into account (six possible values of F1 and two possible decisions). For each of these conditions, we can average each neuron's spike trains over trials and then smooth the resulting time series in order to estimate the neuron's time-dependent firing rate (also known as peri-stimulus time histogram or PSTH). We find that the PSTHs of many neurons are tuned to the stimulus F1, the decision, or both (*Figure 1b*; so-called *mixed selectivity*), and different neurons generally show different tuning. Our goal is to characterize and summarize the tuning of all $N$ recorded neurons.

The most standard and widespread approach is to resort to a statistical test (e.g. a two-way analysis of variance or ANOVA), in order to check whether the firing rate of a neuron depends significantly on the frequency F1 or on the monkey's decision. Such a test can be run for each neuron and each time point, in which case the population tuning over time is often summarized as the fraction of cells significantly tuned to stimulus or decision at each time point ($p<0.05$, *Figure 1c*). In addition to providing such a 'summary statistics', this approach is also used to directly visualize the population activity. For that purpose, one selects the subset of neurons significantly tuned to stimulus or decision (e.g. by focusing on a particular time point, *Figure 1d*) and then averages their PSTHs. The resulting 'population average' is shown in *Figure 1e*, where we also took the sign of the effect size into account. The population average is generally thought to demonstrate the 'most typical' firing pattern among the cells encoding the corresponding parameter. Importantly, this method yields one single population average or 'component' for each parameter. Each such component can be understood as a linear combination (or a linear *readout*) of the individual PSTHs, with all $N_s$ significant neurons for a parameter having the same weights $\pm 1/N_s$ and all others having weight zero.

In a related approach, the firing rates of each neuron at each time point are linearly regressed on stimulus and decision (*Figure 1f*) (*Brody et al., 2003*). *Mante et al. (2013)* suggested to use the regression coefficients of all $N$ neurons (*Figure 1g*) as weights to form linear combinations of PSTHs representing stimulus and decision tuning (*Figure 1h*). This approach, which the authors call 'targeted dimensionality reduction' (TDR), also yields one component per task parameter: in our example, we obtain one component for the stimulus and one for the decision (*Figure 1h*; see Materials and methods for details).

Both of these approaches are *supervised*, meaning that they are informed by the task parameters. At the same time, they do not seek to faithfully represent the whole dataset and are prone to losing some information about the neural activities. Indeed, the two components from *Figure 1e* explain only 23% of the total variance of the population firing rates and the two components from *Figure 1h* explain only 22% (see Materials and methods). Consequently, a naive observer would not be able to infer from the components what the original neural activities looked like.

While such supervised approaches can be extended in various ways to produce more components and capture more variance, a more direct way to avoid this loss of information is to resort to *unsupervised* methods such as principal component analysis (PCA). This method extracts a set of principal components (PCs) that are linear combinations of the original PSTHs, just as the population averages above. However, the weights to form these linear combinations are chosen so as to maximize the amount of explained variance (first six components explain 69% of variance, see *Figure 1i–k*). The principal components can be thought of as 'building blocks' of neural activity: PSTHs of actual neurons are given by linear combinations of PCs, with the first PCs being more informative than the later ones. However, since PCA is an unsupervised method, information about stimuli and decisions is not taken into account, and the resulting components can retain mixed selectivity and therefore fail to highlight neural tuning to the task parameters.

The most striking observation when comparing supervised and unsupervised approaches is how different the results look. Indeed, PCA paints a much more complex picture of the population activity, dominated by strong temporal dynamics, with several stimulus- and decision-related components. At the same time, none of the methods can fully demix the stimulus and decision information: even the supervised methods show decision-related activity in the stimulus components and stimulus-related activity in the decision components (*Figure 1e,h*).

## Demixed principal component analysis (dPCA)

To address these problems, we developed a modified version of PCA that not only compresses the data, but also demixes the dependencies of the population activity on the task parameters. We will

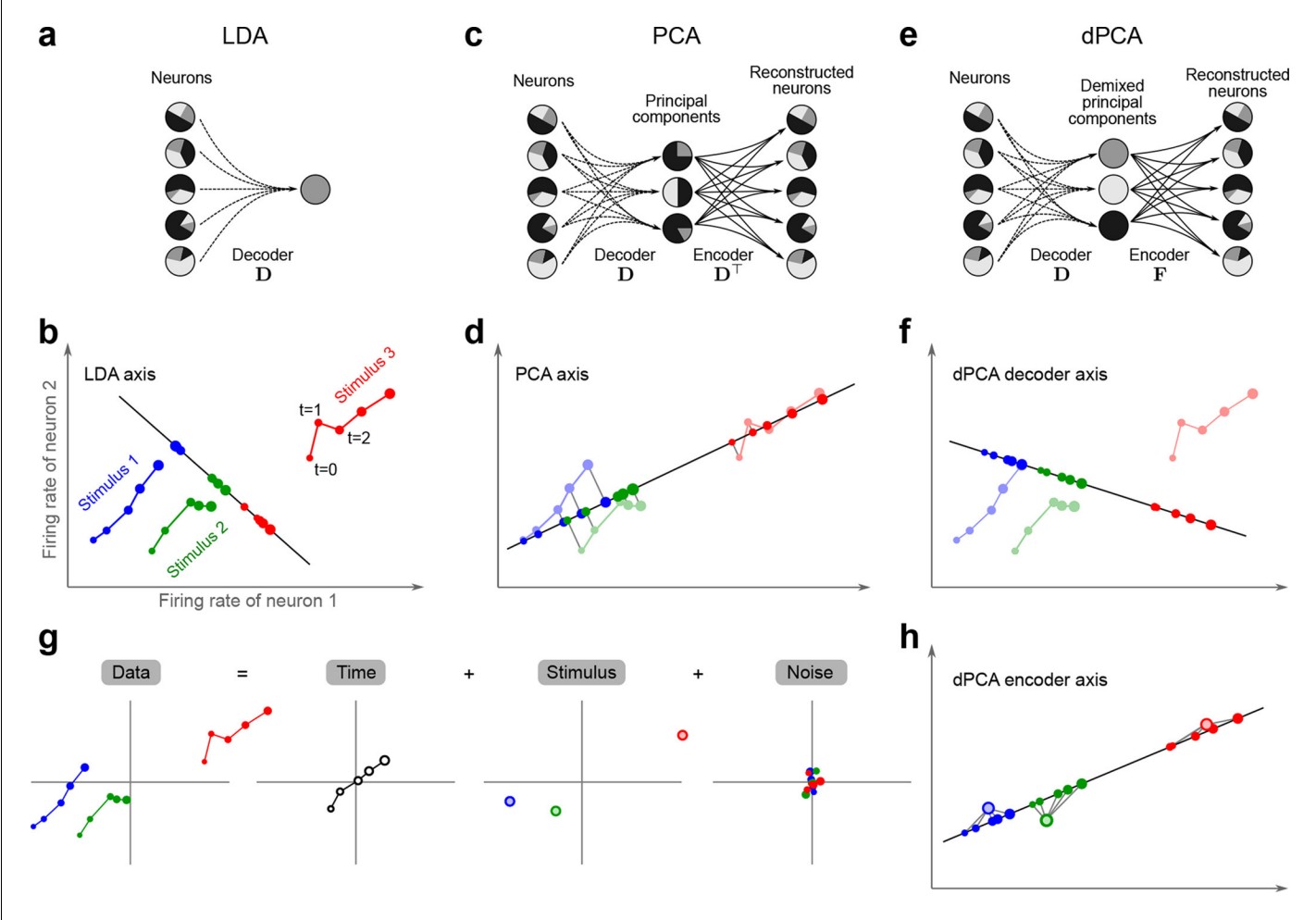

**Figure 2.** Linear dimensionality reduction. (a) Linear discriminant analysis maps the firing rates of individual neurons onto a latent component that allows us to decode a task parameter of interest. Shades of grey inside each neuron show the proportion of variance due to the various task parameters (e.g. stimulus, decision, and time), illustrating mixed selectivity. In contrast, the LDA component is maximally demixed. (b) At any moment in time, the population firing rate of $N$ neurons is represented by a point in the $N$-dimensional space; here $N = 2$. Each trial is represented by a trajectory in this space. Colors indicate different stimuli and dot sizes represent time. The LDA component for stimulus is given by the projection onto the LDA axis (black line); projections of all points are shown along this line. All three stimuli are clearly separated, but their geometrical relation to each other is lost. (c) Principal component analysis linearly maps the firing rates into a few principal components such that a second linear transformation can reconstruct the original firing rates. (d) The same set of points as in (b) is projected onto the first PCA axis. However, the stimuli are no longer separated. Rather, the points along the PCA axis have complex dependencies on stimulus and time (mixed selectivity). The PCA axis minimizes the distances between the original points and their projections. (e) Demixed principal component analysis also compresses and decompresses the firing rates through two linear transformations. However, here the transformations are found by both minimizing the reconstruction error and enforcing a demixing constraint on the latent variables. (f) The same set of points as in (b) projected onto the first dPCA decoder axis. The three stimuli are clearly separated (as in LDA), but some information about the relative distances between classes is preserved as well (as in PCA). (g) The same data as in (b) linearly decomposed into the time effect, the stimulus effect, and the noise. (h) The dPCA projection from (f) has to be mapped onto a different axis, given by the dPCA encoder, in order to reconstruct the stimulus class means (large colored circles). The decoder and encoder axes together minimize the reconstruction error between the original data and the stimulus class means.

first explain that these two goals generally constitute a trade-off, then suggest a solution to this trade-off for a single task parameter, and then generalize to multiple task parameters.

The trade-off between demixing and compression is illustrated in *Figure 2*, where we compare linear discriminant analysis (LDA, *Figure 2a,b*), PCA (*Figure 2c,d*), and dPCA (*Figure 2e–h*). We will first focus on a single task parameter and seek to reduce the activity of $N = 2$ neurons responding to three different stimuli. For each stimulus, the joint activity of the two neurons traces out a trajectory

in the space of firing rates as time progresses (*Figure 2b*). The aim of 'demixing' in this simplified case is to find a linear mapping (*decoder*) of the neural activity that separates the different stimuli (*Figure 2a*) and ignores the time-dependency. We can use LDA in order to determine a projection of the data that optimally separates the three stimuli. However, LDA will generally not preserve the 'geometry' of the original neural activity: firing patterns for stimuli 1 and 2 are close to each other and far away from stimulus 3, whereas in the LDA projection all three stimuli are equally spaced (*Figure 2b*). More generally, decoding is always prone to distorting the data and therefore tends to impede a proper reconstruction of the original data from the reduced description.

The aim of compression is to find a linear mapping (decoder) that reduces the dimensionality and preserves the original data as much as possible (*Figure 2c,d*). Using PCA, we determine a projection of the data that minimizes the reconstruction error between the projections and the original points. In contrast to LDA, PCA seeks to preserve the geometry of the neural activity, and thereby yields the most faithful reduction of the data (*Figure 2d*). However, the PCA projection does not properly separate the stimuli and mixes the time-dependency with the stimulus-dependency.

The wildly different projection axes for LDA (*Figure 2b*) and PCA (*Figure 2d*) seem to suggest that the goals of demixing and compression are essentially incompatible in this example. However, we can achieve both goals by assuming that the reconstruction of the original data works along a separate *encoder* axis (*Figure 2f,h*). Given this additional flexibility, we first choose a decoder axis that reconciles the decoding and compression objectives. Once projected onto this axis, all three stimuli are separated from each other, as in LDA, yet their geometrical arrangement is approximately preserved, as in PCA (*Figure 2f*). In turn, when reconstructed along the encoder axis, the projected data still approximates the original data (*Figure 2h*).

To define these ideas more formally, we assume that we simultaneously recorded the spike trains of $N$ neurons. Let $\mathbf{X}$ be our data matrix with $N$ rows, in which the $i$-th row contains the instantaneous firing rate (i.e. binned or smoothed spike train) of the $i$-th neuron for all task conditions and all trials (assumed to be centered, i.e., with row means subtracted). Classical PCA compresses the data with a decoder matrix $\mathbf{D}$. The resulting principal components can then be linearly de-compressed through an encoder matrix $\mathbf{D}^\top$, approximately reconstructing the original data (*Hastie et al., 2009*). The optimal decoder matrix is found by minimizing the squared error between the original data, $\mathbf{X}$, and the reconstructed data, $\mathbf{D}^\top \mathbf{D} \mathbf{X}$, given by

$$L_{\mathrm{PCA}} = \|\mathbf{X} - \mathbf{D}^\top \mathbf{D} \mathbf{X}\|^2.$$

In the toy example of *Figure 2*, the data matrix $\mathbf{X}$ is of size $2 \times 15$, and the decoder matrix $\mathbf{D}$ is of size $1 \times 2$. Crucially, the information about task parameters does not enter the loss function and hence PCA neither decodes nor demixes these parameters.

In our method, which we call *demixed PCA (dPCA)*, we make two changes to this classical formulation. First, we require that the compression and decompression steps reconstruct not the neural activity directly, but the neural activity averaged over trials and over some of the task parameters. In the toy example, the reconstruction target is the matrix of stimulus averages, $\mathbf{X}_s$, which has the same size as $\mathbf{X}$, but in which every data point is replaced by the average neural activity for the corresponding stimulus, as shown in *Figure 2h*. Second, we gain additional flexibility in this quest by compressing the data with a linear mapping $\mathbf{D}$, yet decompressing it with another linear mapping $\mathbf{F}$ (*Figure 2e*). The respective matrices are chosen by minimizing the loss function

$$L_{\mathrm{dPCA}} = \|\mathbf{X}_s - \mathbf{F} \mathbf{D} \mathbf{X}\|^2.$$

Accordingly, for each stimulus, the neural activities are projected close to the average stimulus, which allows us both to decode the stimulus value and to preserve the relative distances of the neural activities.

In order to see how this approach preserves all aspects of the original data, and not just some averages, we note that the data in our toy example included both stimulus and time. The matrix $\mathbf{X}_s$ can be understood as part of a linear decomposition of the full data $\mathbf{X}$ into parameter-specific averages: a time-varying part, $\mathbf{X}_t$, that is obtained by averaging $\mathbf{X}$ over stimuli, and a stimulus-varying part, $\mathbf{X}_s$, that is obtained by averaging $\mathbf{X}$ over time. Any remaining parts of the activity are captured in a noise term (*Figure 2g*). In turn, we can find separate decoder and encoder axes for each of these averages. Once more than $N = 2$ neurons are considered, these decoder and encoder axes

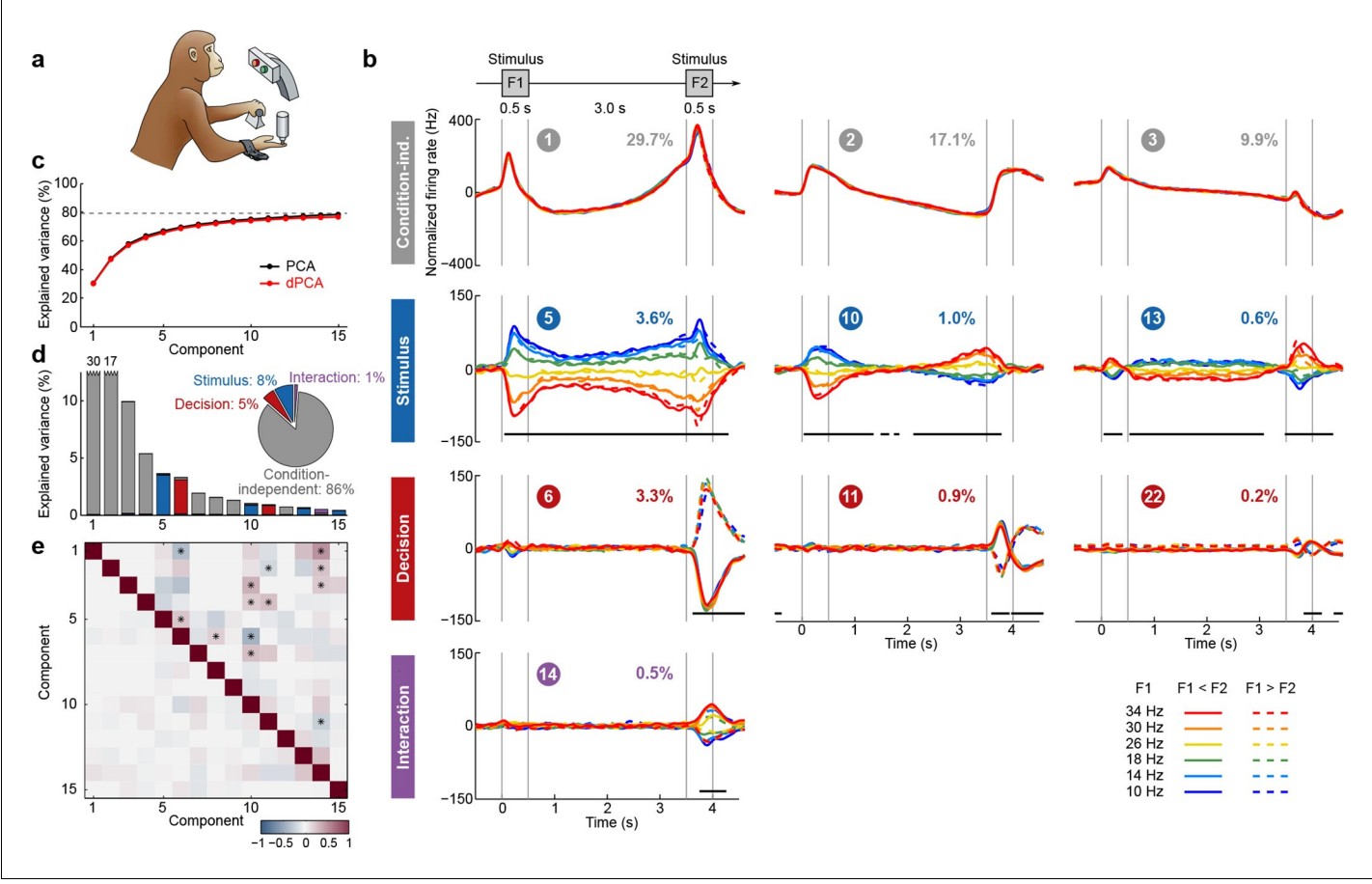

**Figure 3.** Demixed PCA applied to recordings from monkey PFC during a somatosensory working memory task (*Romo et al., 1999*). (a) Cartoon of the paradigm, adapted from *Romo and Salinas (2003)*. (b) Demixed principal components. Top row: first three condition-independent components; second row: first three stimulus components; third row: first three decision components; last row: first stimulus/decision interaction component. In each subplot, the full data are projected onto the respective dPCA decoder axis, so that there are 12 lines corresponding to 12 conditions (see legend). Thick black lines show time intervals during which the respective task parameters can be reliably extracted from single-trial activity (using pseudotrials with all recorded neurons), see Materials and methods. Note that the vertical scale differs across rows. Ordinal number of each component is shown in a circle; explained variances are shown as percentages. (c) Cumulative variance explained by PCA (black) and dPCA (red). Demixed PCA explains almost the same amount of variance as standard PCA. Dashed line shows an estimate of the fraction of 'signal variance' in the data, the remaining variance is due to noise in the PSTH estimates (see Materials and methods). (d) Variance of the individual demixed principal components. Each bar shows the proportion of total variance, and is composed out of four stacked bars of different color: gray for condition-independent variance, blue for stimulus variance, red for decision variance, and purple for variance due to stimulus-decision interactions. Each bar appears to be single-colored, which signifies nearly perfect demixing. Pie chart shows how the total signal variance is split among parameters. (e) Upper-right triangle shows dot products between all pairs of the first 15 demixed principal axes. Stars mark the pairs that are significantly and robustly non-orthogonal (see Materials and methods). Bottom-left triangle shows correlations between all pairs of the first 15 demixed principal components. Most of the correlations are close to zero.

constitute a dimensionality reduction step that reduces the data into a few components, each of which properly decodes one of the task parameters. In turn, the original neural activity can be reconstructed through linear combinations of these components, just as in PCA.

The key ideas of this toy example can be extended to any number of task parameters. In this manuscript, all datasets will have three parameters: time, stimulus, and decision, and we will decompose the neural activities into five parts: condition-independent, stimulus-dependent, decision-dependent, dependent on the stimulus-decision interaction, and noise (see *Figure 8* in the Materials and methods):

$$\mathbf{X} = \mathbf{X}_t + \mathbf{X}_{st} + \mathbf{X}_{dt} + \mathbf{X}_{sdt} + \mathbf{X}_{\text{noise}} = \sum_\phi \mathbf{X}_\phi + \mathbf{X}_{\text{noise}}.$$

Individual terms are again given by a series of averages. This decomposition is fully analogous to the variance (covariance) decomposition done in ANOVA (MANOVA). The only important difference is that the standard (M)ANOVA decomposition for three parameters A, B, and C, would normally have $2^3 = 8$ terms corresponding to the main effects of A, B, C, pairwise interactions AB, BC, and AC, three-way interaction ABC, and the noise. Here we join some of these terms together, as we are not interested in demixing those (see Materials and methods).

Once this decomposition is performed, dPCA finds separate decoder and encoder matrices for each term $\phi$ by minimizing the loss function

$$L_{\text{dPCA}} = \sum_\phi \|\mathbf{X}_\phi - \mathbf{F}_\phi \mathbf{D}_\phi \mathbf{X}\|^2.$$

Each term within the sum can be minimized separately by using *reduced-rank regression*, the solution of which can be obtained analytically in terms of singular value decompositions (see Materials and methods). Each row $\mathbf{d}$ of each $\mathbf{D}_\phi$ yields one *demixed principal component* $\mathbf{dX}$ and, similar to PCA, we order the components by the amount of explained variance. Note that the decoder/encoder axes corresponding to two different task parameters $\phi_1$ and $\phi_2$ are found independently from each other and may end up being non-orthogonal (in contrast to PCA where principal axes are all orthogonal). In a nutshell, the loss function ensures that each set of decoder/encoder axes reconstructs the individual, parameter-specific terms, $\mathbf{X}_\phi$, thereby yielding proper demixing, and the data decomposition ensures that the combination of all decoder/encoder pairs allows to reconstruct the original data, $\mathbf{X}$.

There are a few other technical subtleties (see Materials and methods for details). (1) We formulated dPCA for simultaneously recorded neural activities. However, all datasets analyzed in this manuscript have been recorded sequentially across many sessions, and so to apply dPCA we have to use 'pseudo-trials'. (2) Similar to any other decoding method, dPCA is prone to overfitting and so we introduce a regularization term and perform cross-validation to choose the regularization parameter. (3) The data and variance decompositions from above are exact only if the dataset is *balanced*, i.e., if the same number of trials were recorded in each condition. If this is not the case, one can use a re-balancing procedure. (4) A previous version of dPCA (*Brendel et al., 2011*) used the same variance decomposition but a different and less flexible loss function. The differences are layed out in the Materials and methods section.

## Somatosensory working memory task in monkey PFC

We first applied dPCA to the dataset presented above (*Romo et al., 1999*; *Brody et al., 2003*), encompassing 832 neurons from two animals. As is typical for PFC, each neuron has a distinct response pattern and many neurons show mixed selectivity (some examples are shown in *Figure 1b*). Several previous studies have sought to make sense of these heterogeneous response patterns by separately analyzing different task periods, such as the stimulation and delay periods (*Romo et al., 1999*; *Brody et al., 2003*; *Machens et al., 2010*; *Barak et al., 2010*), the decision period (*Jun et al., 2010*), or both (*Hernández et al., 2010*). With dPCA, however, we can summarize the main features of the neural activity across the whole trial in a single figure (*Figure 3*).

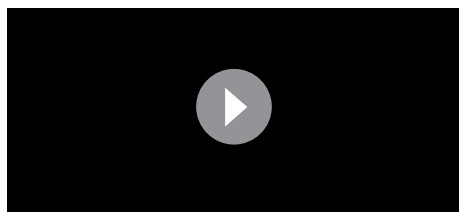

**Video 1.** Stimulus representation in the somatosensory working memory task Two leading stimulus dPCs in the somatosensory working memory task (components #5 and #10 as horizontal and vertical axis correspondingly). Each frame of this movie corresponds to one time point $t$. Each dot is the average between two decision conditions with the same F1 stimulus. Fading 'tails' show last sections of the trajectories. See *Figure 3* for the color code.

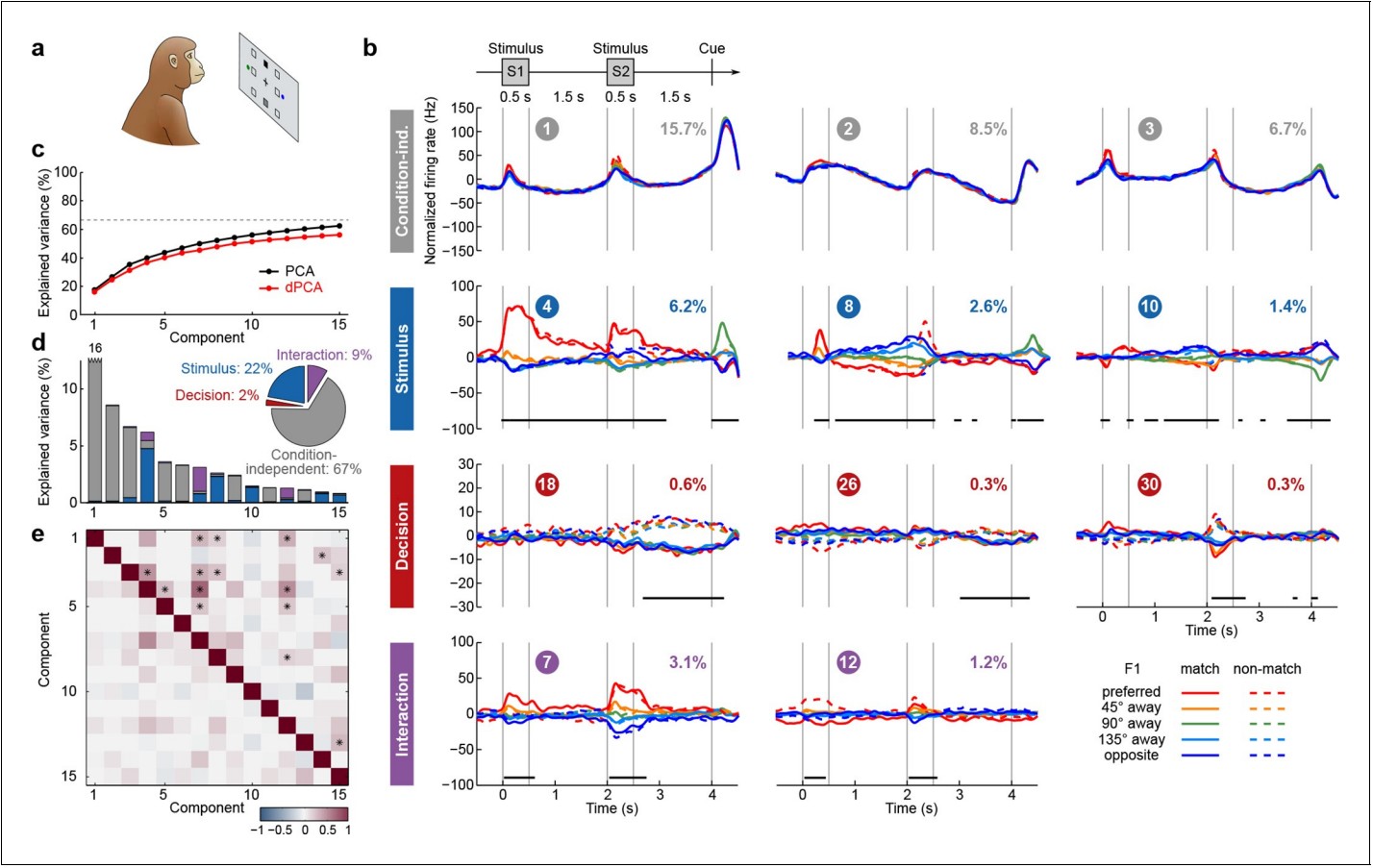

**Figure 4.** Demixed PCA applied to recordings from monkey PFC during a visuospatial working memory task (*Qi et al., 2011*). Same format as *Figure 3*. (a) Cartoon of the paradigm, adapted from *Romo and Salinas (2003)*. (b) Demixed principal components. In each subplot there are ten lines corresponding to ten conditions (see legend). Color corresponds to the position of the last shown stimulus (first stimulus for $t < 2$ s, second stimulus for $t > 2$ s). In non-match conditions (dashed lines) the colour changes at $t = 2$ s. Solid lines correspond to match conditions and do not change colors. (c) Cumulative variance explained by PCA and dPCA components. Dashed line marks fraction of signal variance. (d) Explained variance of the individual demixed principal components. Pie chart shows how the total signal variance is split between parameters. (e) Upper-right triangle shows dot products between all pairs of the first 15 demixed principal axes, bottom-left triangle shows correlations between all pairs of the first 15 demixed principal components.

Just as in PCA, we can think of the demixed principal components (*Figure 3b*) as the 'building blocks' of the observed neural activity, in that the activity of each single neuron is a linear combination (weighted average) of these components. These building blocks come in four distinct categories: some are condition-independent (*Figure 3b*, top row); some depend only on stimulus F1 (second row); some depend only on decision (third row); and some depend on stimulus and decision together (bottom row). The components can be easily seen to demix the parameter dependencies, which is exactly what dPCA aimed for. Indeed, the components shown in *Figure 3b* are projections of the PSTHs of all neurons onto the most prominent decoding axes; each projection (each subplot) shows 12 lines corresponding to 12 conditions. As intended, condition-independent components have all 12 lines closely overlapping, stimulus components have two lines for each stimulus closely overlapping, etc.

The overall variance explained by the dPCA components (*Figure 3c*, red line) is very close to the overall variance explained by the PCA components (black line). Accordingly, we barely lost any variance by imposing the demixing constraint, and the population activity is accurately represented by the obtained dPCA components.

The dPCA analysis captures the major findings previously obtained with these data: the persistence of the F1 tuning during the delay period (component #5; *Romo et al., 1999*; *Machens et al.,*

2005), the temporal dynamics of short-term memory (components ##5, 10, 13; *Brody et al., 2003*; *Machens et al., 2010*; *Barak et al., 2010*), the 'ramping' or 'climbing' activities in the delay period (components ##1–3; *Brody et al., 2003*; *Machens et al., 2010*); and pronounced decision-related activities (component #6, *Jun et al., 2010*). We note that the decision components resemble derivatives of each other; these higher-order derivatives likely arise due to slight variations in the timing of responses across neurons (see Appendix B for more details).

The first stimulus component (#5) looks similar to the stimulus components that we obtained with standard regression-based methods (*Figure 1e,h*) but now we have further components as well. Together they show how stimulus representation evolves in time. In particular, plotting the first two stimulus components against each other (see *Video 1*) illustrates how stimulus representation rotates in the neural space during the delay period so that the encoding subspaces during F1 and F2 periods are not the same (but far from orthogonal either).

As explained above, the demixed principal axes are not constrained to be orthogonal. The angles between the encoding axes are shown in *Figure 3e*, upper-right triangle; we discuss them later, together with other datasets. Pairwise correlations between components are all close to zero (*Figure 3e*, lower-left triangle), as should be expected since the components are considered to represent independent signals.

To assess whether the condition tuning of individual dPCA components was statistically significant, we used each component as a linear decoder to classify conditions. Specifically, stimulus components were used to classify stimuli, decision components to classify decisions, and interaction components to classify all 12 conditions. We used cross-validation to measure time-dependent classification accuracy and a shuffling procedure to assess whether it was significantly above chance (see Materials and methods). Time periods of significant tuning are marked in *Figure 3b* with horizontal black lines.

## Visuospatial working memory task in monkey PFC

We next applied dPCA to recordings from the PFC of monkeys performing a visuospatial working memory task (*Qi et al., 2011*, *2012*; *Meyer et al., 2011*). In this task, monkeys first fixated a small white square at the centre of a screen, after which a square S1 appeared for 0.5 s in one of eight locations around the centre (*Figure 4a*). After a 1.5 s delay, a second square S2 appeared for 0.5 s in either the same ('match') or the opposite ('non-match') location. Following another 1.5 s delay, a green and a blue choice target appeared in locations orthogonal to the earlier presented stimuli. Monkeys had to saccade to the green target to report a match condition, and to the blue one to report a non-match.

We analyzed the activity of 956 neurons recorded in the lateral PFC of two monkeys performing this task. Proceeding exactly as before, we obtained the average time-dependent firing rate of each neuron for each condition. Following the original studies, we eliminated the trivial rotational symmetry of the task by collapsing the eight possible stimulus locations into five locations that are defined with respect to the preferred location of each neuron (0°, 45°, 90°, 135°, or 180° away from the preferred location, see Materials and methods). As a consequence, we obtained ten conditions: five possible stimulus locations, each paired with two possible decisions of the monkey.

The dPCA results are shown in *Figure 4*. As before, stimulus and decision are well separated at the population level despite being intermingled at the single-neuron level; at the same time dPCA captures almost the same amount of variance as PCA. One notable difference from before is the presence of strong interaction components in *Figure 4b*. However, these interaction components are in fact stimulus components in disguise. In match trials, S2 and S1 appear at the same location, and in non-match trials at opposite locations. Information about S2 is therefore given by a non-linear function of stimulus S1 and the trial type (i.e. decision), which is here captured by the interaction components.

Here again, our analysis summarizes previous findings obtained with this dataset. For instance, the first and the second decision components show tuning to the match/non-match decision during the S2 period and in the subsequent delay period. Using these components as fixed linear decoders, we achieve single-trial classification accuracy of match vs. non-match of 75% for $t > 2$ (cross-validated, see Materials and methods, *Figure 12*), which is approximately equal to the state-of-the-art classification performance reported previously (*Meyers et al., 2012*).

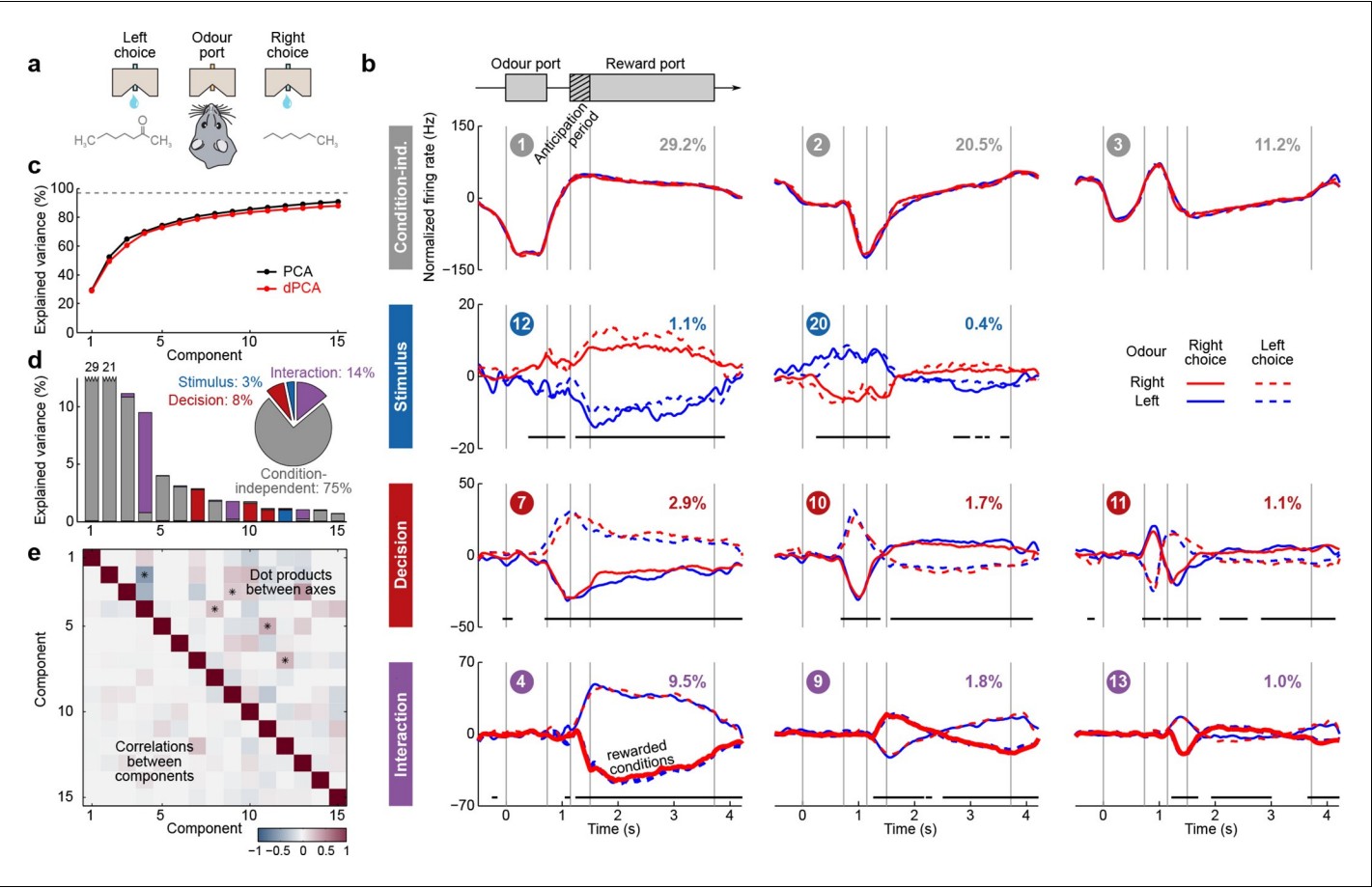

**Figure 5.** Demixed PCA applied to recordings from rat OFC during an olfactory discrimination task (*Feierstein et al., 2006*). Same format as *Figure 3*. (a) Cartoon of the paradigm, adapted from *Wang et al. (2013)*. (b) Each subplot shows one demixed principal component. In each subplot there are four lines corresponding to four conditions (see legend). Two out of these four conditions were rewarded and are shown by thick lines. (c) Cumulative variance explained by PCA and dPCA components. (d) Explained variance of the individual demixed principal components. Pie chart shows how the total signal variance is split between parameters. (e) Upper-right triangle shows dot products between all pairs of the first 15 demixed principal axes, bottom-left triangle shows correlations between all pairs of the first 15 demixed principal components.

Constantinidis et al. have also recorded population activity in PFC before starting the training (both S1 and S2 stimuli were presented exactly as above, but there were no cues displayed and no decision required). When analyzing this pre-training population activity with dPCA, the first stimulus and the first interaction components come out close to the ones shown in *Figure 4*, but there are no decision and no 'memory' components present (data not shown), in line with previous findings (*Meyers et al., 2012*). These task-specific components appear in the population activity only after extensive training.

## Olfactory discrimination task in rat OFC

Next, we applied dPCA to recordings from the OFC of rats performing an odor discrimination task (*Feierstein et al., 2006*). This behavioral task differs in two crucial aspects from the previously considered tasks: it requires no active storage of a stimulus, and it is self-paced. To start a trial, rats entered an odor port, which triggered delivery of an odor with a random delay of 0.2–0.5 s. Each odor was uniquely associated with one of the two available water ports, located to the left and to the right from the odor port (*Figure 5a*). Rats could sample the odor for as long as they wanted (up to 1 s), and then had to move to one of the water ports. If they chose the correct water port, reward was delivered following an anticipation period of random length (0.2–0.5 s).

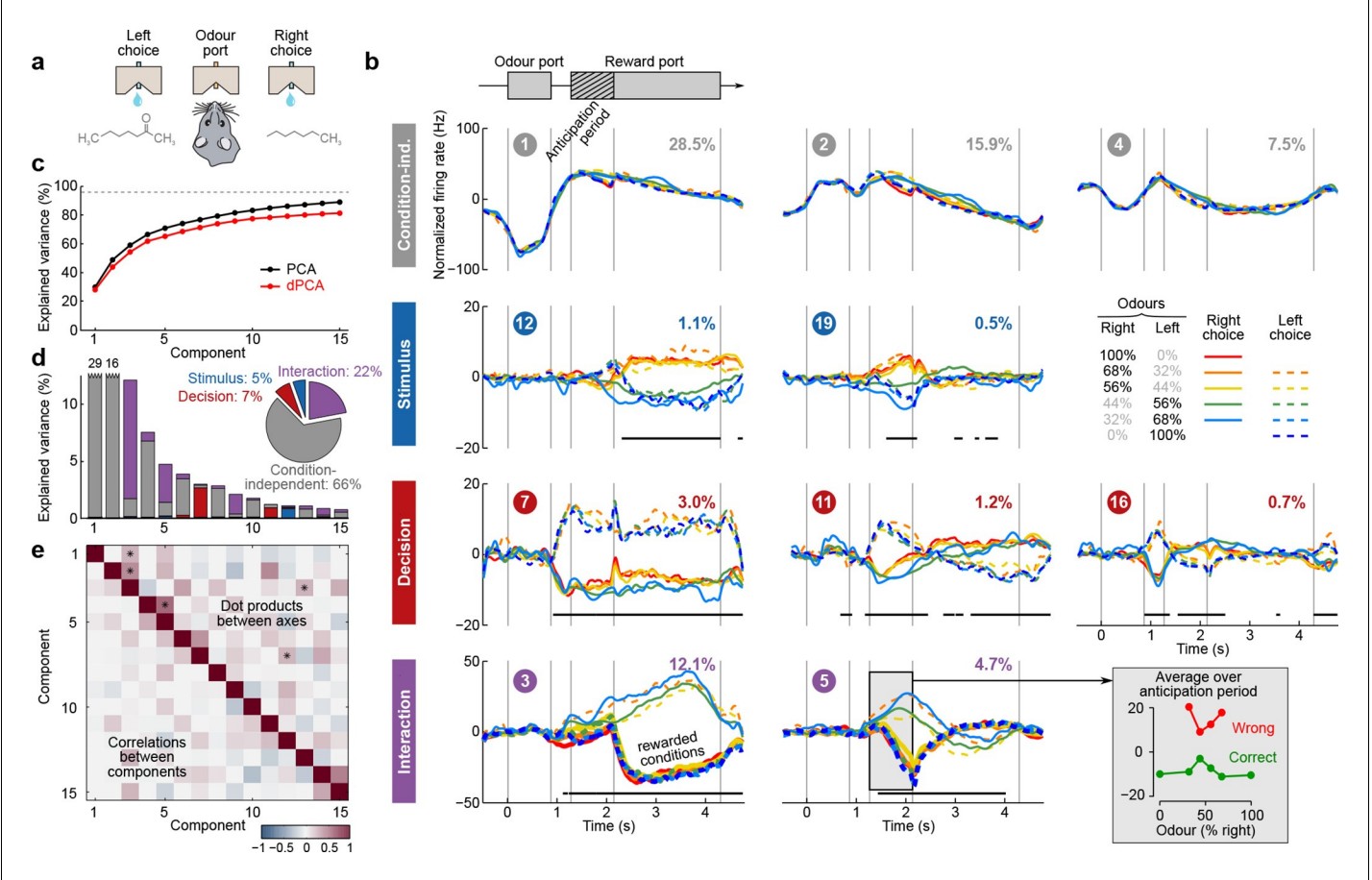

**Figure 6.** Demixed PCA applied to recordings from rat OFC during an olfactory categorization task (*Kepecs et al., 2008*). Same format as *Figure 3* (a) Cartoon of the paradigm, adapted from *Wang et al. (2013)*. (b) Each subplot shows one demixed principal component. In each subplot there are ten lines corresponding to ten conditions (see legend). Six out of these ten conditions were rewarded and are shown with thick lines; note that the pure left (red) and the pure right (blue) odors did not have error trials. Inset shows mean rate of the second interaction component during the anticipation period. (c) Cumulative variance explained by PCA and dPCA components. (d) Explained variance of the individual demixed principal components. Pie chart shows how the total signal variance is split between parameters. (e) Upper-right triangle shows dot products between all pairs of the first 15 demixed principal axes, bottom-left triangle shows correlations between all pairs of the first 15 demixed principal components.

We analyzed the activity of 437 neurons recorded in five rats in four conditions: two stimuli (left and right) each paired with two decisions (left and right). Two of these conditions correspond to correct (rewarded) trials, and two correspond to error (unrewarded) trials. Since the task was self-paced, each trial had a different length; in order to align events across trials, we restretched (time-warped) the firing rates in each trial (see Materials and methods). Alignment methods without time warping led to similar results (data not shown).

Just as neurons from monkey PFC, neurons in rat OFC exhibit diverse firing patterns and mixed selectivity (*Feierstein et al., 2006*). Nonetheless, dPCA was able to demix the population activity (*Figure 5*). In this dataset, interaction components separate rewarded and unrewarded conditions (thick and thin lines in *Figure 5b*, bottom row), i.e., correspond to neurons tuned either to reward, or to the absence of reward.

The overall pattern of neural tuning across task epochs agrees with the findings of the original study (*Feierstein et al., 2006*). Interaction components are by far the most prominent among all the condition-dependent components, corresponding to the observation that many neurons are tuned to the presence/absence of reward. Decision components come next, with the caveat that decision information may also reflect the rat's movement direction and/or position, as was pointed out previously (*Feierstein et al., 2006*). Stimulus components are less prominent, but nevertheless show clear

stimulus tuning, demonstrating that even in error trials there is reliable information about stimulus identity in the population activity.

Curiously, the first interaction component (#4) already shows significant tuning to reward in the anticipation period. In other words, neurons tuned to presence/absence of reward start firing before the reward delivery (or, on error trials, before the reward could have been delivered). We return to this observation in the next section.

## Olfactory categorization task in rat OFC

*Kepecs et al. (2008)* extended the experiment of *Feierstein et al. (2006)* by using odor mixtures instead of pure odors, thereby varying the difficulty of each trial (*Uchida and Mainen, 2003*). In each trial, rats experienced mixtures of two fixed odors with different proportions (*Figure 6a*). Left choices were rewarded if the proportion of the 'left' odor was above 50%, and right choices otherwise. Furthermore, the waiting time until reward delivery (anticipation period) was increased to 0.3–2 s.

We analyzed the activity of 214 OFC neurons from three rats recorded in 8 conditions, corresponding to four odor mixtures, each paired with two decisions (left and right). During the presentation of pure odors (100% right and 100% left) rats made essentially no mistakes, and so we excluded these data from the dPCA computations (which require that all parameter combinations are present, see Discussion). Nevertheless, we displayed these additional two conditions in *Figure 6*.

The dPCA components shown in *Figure 6b* are similar to those presented in *Figure 5b*. Here again, some of the interaction components (especially the second one, #5) show strong tuning already during the anticipation period, i.e. before the actual reward delivery. The inset in *Figure 6b* shows the mean value of the component #5 during the anticipation period, separating correct (green) and incorrect (red) trials for each stimulus. The characteristic U-shape for the error trials and the inverted U-shape for the correct trials agrees well with the predicted value of the rat's uncertainty in each condition (*Kepecs et al., 2008*). Accordingly, this component can be interpreted as corresponding to the rat's uncertainty or confidence about its own choice, confirming the results of *Kepecs et al. (2008)*. In summary, both the main features of this dataset, as well as some of the subtleties, are picked up and reproduced by dPCA.

## Universal features of the PFC population activity

One of the key advantages of applying dPCA to these four datasets is that we can now compare them far more easily than was previously possible. This comparison allows us to highlight several general features of the population activity in prefrontal areas.

First, most of the variance of the neural activity is always captured by the condition-independent components that together amount to 65–90% of the signal variance (see pie charts in *Figures 3–6d*; see Materials and methods for definition of 'signal variance'). These components capture the temporal modulations of the neural activity throughout the trial, irrespective of the task condition. Their striking dominance in the data may come as a surprise, as such condition-independent components are usually not analyzed or shown (cf. *Figure 1e,h*), even though condition-independent firing has been described even in sensory areas (*Sornborger et al., 2005*). These components are likely explained in part by an overall firing rate increase during certain task periods (e.g. during stimulus presentation). More speculatively, they could also be influenced by residual sensory or motor variables that vary rhythmically with the task, but are not controlled or monitored (*Renart and Machens, 2014*). The attentional or motivational state of animals, for instance, often correlates with breathing (*Huijbers et al., 2014*), pupil dilation (*Eldar et al., 2013*), body movements (*Gouvêa et al., 2014*), etc.

Second, even though dPCA, unlike PCA, does not enforce orthogonality between encoding axes corresponding to different task parameters, most of them turned out to be close to orthogonal to each other (*Figures 3–6e*, upper triangle), as has been observed before (*Brendel et al., 2011*; *Rishel et al., 2013*; *Raposo et al., 2014*). Nevertheless, many pairs were significantly non-orthogonal, meaning that neurons expressing one of the components tended to also express the other one. Throughout the four datasets, we identified 277 pairs of axes (among the first 15 axes) corresponding to different parameters. Of these, 38, i.e. 14%, were significantly non-orthogonal with $p<0.001$ (8 out of 53 if we do not take time axes into account).

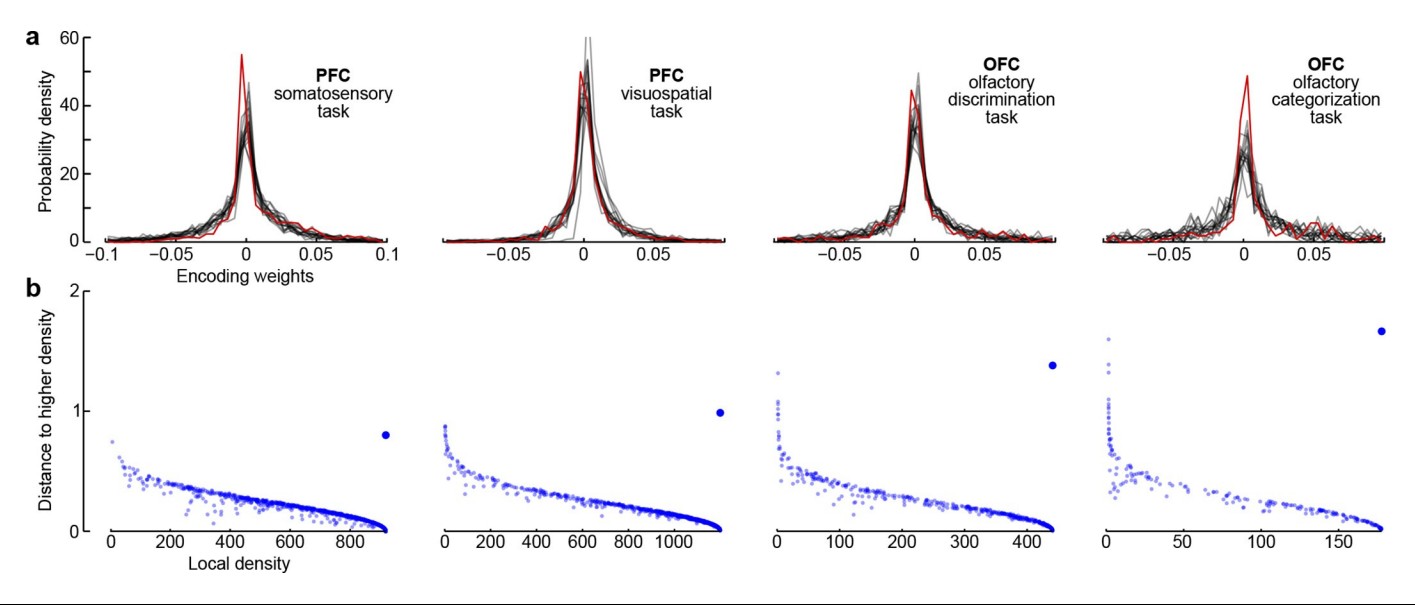

**Figure 7.** Encoder weights for the leading dPCA components across the neural population. (a) Distributions of encoder weights for the 15 leading dPCA components across the neural population, in each of the four datasets. Each subplot shows 15 probability density curves, one curve per component (bin width 0.005). The distribution corresponding to the first component is highlighted in red. (b) Clustering of neurons by density peaks (*Rodriguez and Laio, 2014*). For each dataset we took the first 15 dPCA components, and then ran the clustering algorithm in the 15-dimensional space of encoding weights. The clustering algorithm works in two steps: first, it computes a local density for each point (i.e., for each neuron), using a Gaussian kernel with $\sigma^2 = 0.01$. Second, for each point it finds the minimal distance to a point with higher local density (if there is no such point, then the distance to the furthest point is taken). Each subplot shows local density on the horizontal axis plotted against distance to the next point with higher density on the vertical axis; each dot corresponds to one of the $N$ neurons. Cluster centres are characterized by high local density and large distance to the point of even higher density; they should appear as outliers in the upper-right corner of the plot (see *Rodriguez and Laio, 2014*, for details). In each case, there is only one such outlier (bigger dot), indicating a single cluster.

Third, all dPCA components in each of the datasets are distributed across the whole neural population (as opposed to being exhibited only by a subset of cells). For each component and each neuron, the corresponding encoder weight shows how much this particular component is exhibited by this particular neuron. For each component, the distribution of weights is strongly unimodal, centred at zero (*Figure 7a*), and rather symmetric (although it is skewed to one side for some components). In other words, there are no distinct sub-populations of neurons predominantly expressing a particular component; rather, each individual neuron can be visualized as a random linear combination of these components. We confirmed this observation by applying a recently developed clustering algorithm (*Rodriguez and Laio, 2014*) to the population of neurons in the 15-dimensional space of dPC weights. In all cases, the algorithm found only one cluster (*Figure 7b*). An alternative clustering analysis with Gaussian mixture models yielded similar results (data not shown). This absence of any detectable clusters of neurons has been noted before (*Machens et al., 2010*) and was recently observed in other datasets as well (*Raposo et al., 2014*).

## Discussion

Mixed selectivity of neurons in higher cortical areas has been increasingly recognized as a problem for the analysis of neurophysiological recordings, with many different approaches suggested to deal with it (*Brody et al., 2003*; *Machens et al., 2010*; *Machens, 2010*; *Brendel et al., 2011*; *Rigotti et al., 2013*; *Pagan and Rust, 2014*; *Park et al., 2014*; *Raposo et al., 2014*; *Cunningham and Yu, 2014*). The main strength and the main novelty of the method suggested here (dPCA) is that it offers a unified and principled way of analyzing such data.

Demixed PCA combines the strengths of existing supervised and unsupervised approaches to neural population data analysis (*Table 1*, see also the first section of the Results). Supervised

**Table 1.** Demixed PCA in comparison with existing methods. Columns: 'Signif.' refers to the method of counting significantly tuned cells, as shown in *Figure 1c–e*. TDR refers to the 'targeted dimensionality reduction' of *Mante et al. (2013)* shown in *Figure 1f–h*. LDA stands for linear discriminant analysis, but this column applies to any classification method (e.g. support vector machine, ordinal logistic regression, etc.). All classification methods can be used to summarize population tuning via a time-dependent classification accuracy (e.g. *Meyers et al., 2012*). PCA stands for principal component analysis, as shown in *Figure 1i–k*. FA stands for factor analysis, GPFA for Gaussian process factor analysis (*Yu et al., 2009*), LDS for hidden linear dynamical system (*Buesing et al., 2012a*; *2012b*), jPCA is the method introduced in *Churchland et al. (2012)* . Some of the existing methods can be extended to become more general, but here we refer to how these methods are actually used in the original research. **Rows:** The first two rows are the two defining goals of dPCA. Following rows highlight notable features of other methods.

| | Signif. | TDR | LDA | PCA | FA | GPFA | jPCA | LDS | dPCA |
|---|---|---|---|---|---|---|---|---|---|
| Takes task parameters into account & provides summary statistics of population tuning | ✓ | ✓ | ✓ | | | | | | ✓ |
| Allows to reconstruct neural firing (captures variance) | | | | ✓ | ✓ | ✓ | | ✓ | ✓ |
| Based on dynamical model | | | | | | | ✓ | ✓ | |
| Based on probabilistic model | | | | | ✓ | ✓ | | ✓ | |
| Takes spike trains as input | | | | | | ✓ | | ✓ | |

methods can characterize population tuning to various parameters of interest but often do not faithfully represent the whole dataset. Unsupervised methods can capture the overall variance but are not informed by task parameters. Our method yields components that capture almost as much variance as PCA does, but are demixed.

We view both properties as equally important. On one hand, demixing can greatly simplify visualization and interpretation of neural population data. Indeed, in all cases presented here, all the major aspects of the population activity that had previously been reported are directly visible on the dPCA summary figure. On the other hand, faithful representation of the population activity (i.e. 'capturing variance') avoids that a particular interpretation distorts characteristic features of the data. The latter feature is particularly important for the development of theoretical models, which otherwise may inherit an interpretation bias without being aware of it.

Apart from being a useful tool for analyzing any particular dataset, dPCA highlights common features of neural activity when applied to several datasets, allowing to adopt a comparative approach to study population activity.

## Relationship to other methods, including our earlier work

The method presented here is conceptually based on our previous work (*Machens, 2010*; *Machens et al., 2010*; *Brendel et al., 2011*), but is technically very different. The original approach from *Machens et al. (2010)* only works for two parameters of interest, such as time and stimulus. *Machens (2010)* suggested a partial generalization to multiple parameters and *Brendel et al. (2011)* introduced the full covariance decomposition and developed a probabilistic model. However, all of them imposed orthogonality on the decoder/encoder axes (and as a result did not distinguish them), a constraint that cannot be easily relaxed. While we have previously argued that orthogonality is a desirable feature of the decomposition, we now believe that it is better not to impose it upfront. First, by looking across many datasets, we have learnt that encoding subspaces can sometimes be highly non-orthogonal (*Figures 3–6e*) and hence not demixable under orthogonality constraints. Second, by not imposing orthogonality, we can easier identify components that are truly orthogonal. Third, removing the orthogonality constraint allowed us to obtain a simple analytical solution in terms of singular value decompositions (see Materials and methods) and hence to avoid local minima, convergence issues, and any additional optimization-related hyperparameters.

To demonstrate these advantages, we ran the algorithm of *Brendel et al. (2011)*, dPCA-2011, on all our datasets. The resulting components were similar to the components presented here, with the amount of variance captured by the first 15 components being very close; but the achieved demixing was worse. For each component we defined a *demixing index* (see Materials and methods) that is equal to 1 if the component is perfectly demixed. For all datasets, these indices were significantly

higher with our current dPCA-2015 method than with dPCA-2011. Moreover, dPCA-2011 failed to find some weak components at all. For comparison, see *Figure 14* in the Materials and methods.

Another method, called 'targeted dimensionality reduction' (TDR) has recently been suggested for neural data analysis and is similar in spirit to dPCA in that it looks for demixing linear projections (*Mante et al., 2013*). As mentioned above, the original application of this method yields only one component per task parameter and ignores the condition-independent components. While TDR can be extended in various ways to yield more components, no principled way of doing it has been suggested so far. Comparison of dPCA with TDR on our datasets shows that dPCA demixes the task-parameter dependencies better than TDR (see *Figure 14* in the Materials and methods).

For an in-depth discussion of the relationship between dPCA and LDA/MANOVA, we refer the reader to the Methods. Briefly, LDA is a one-way technique, meaning that only one parameter (class id) is associated with each data point. Therefore, LDA cannot directly be applied to the demixing problem. While LDA could be generalized to deal with several parameters in a systematic way, such a generalization has not been used for dimensionality reduction of neural data and does not have an established name in the statistical literature (we call it *factorial LDA*). We believe that for the purposes of dimensionality reduction, dPCA is a superior approach since it combines a reasonably high class separation with low reconstruction error, whereas LDA only optimizes class separation without taking the (potential) reconstruction error into account (see *Figure 2*). MANOVA, on the other hand, is a statistical test closely related to LDA that deals with multiple parameters. However, it deals with isolating the contribution of each parameter from residual noise rather than from the other parameters, and is therefore not suited for demixing.

## Limitations and future work

While we believe that dPCA is an easy-to-use method of visualizing complex data sets with multiple task parameters, several limitations should be kept in mind. First, dPCA as presented here works only with discrete parameters, and all possible parameter combinations must be present in the data. This limitation is the downside of the large flexibility of the method: apart from the demixing constraint, we do not impose any other constraints on the latent variables and their estimation remains essentially non-parametric. In order to be able to treat continuous parameters or missing data (missing parameter combinations), we would need to further constrain the estimation of these latent variables, using e.g. a parametric model. One simple possibility is to directly use a parametric model for the activity of the single neurons, such as the linear model used in *Mante et al. (2013)*, in order to fill in any missing data points, and then run dPCA subsequently.

Second, the number of neurons needs to be sufficiently high in order to obtain reliable estimates of the demixed components. In our datasets, we found that at least $\sim 100$ neurons were needed to achieve satisfactory demixing. The number is likely to be higher if more than three task parameters are to be demixed, as the number of interaction terms grows exponentially with the number of parameters. This trade-off between model complexity and demixing feasibility should be kept in mind when deciding how many parameters to put into the dPCA procedure. In cases when there are many task parameters of interest, dPCA is likely to be less useful than the more standard parametric single-unit approaches (such as linear regression). As a trivial example, imagine that only $N = 1$ neuron has been recorded; it might have strong and significant tuning to various parameters of interest, but there is no way to demix (or decode) these parameters from the recorded 'population.'

Third, even with a large number of neurons, a dataset may be non-demixable, in which case dPCA would fail. For instance, if the high-variance directions of the stimulus and the decision parts of the neural activities fully overlap, then there is no linear decoder that can demix the two parameters.

Finally, dPCA components corresponding to the same parameter (e.g. successive stimulus components) are here chosen to be orthogonal, similarly to PCA. This can make successive components difficult to interpret (e.g. the second and the third stimulus components in *Figure 3*). To make them more interpretable, the orthogonality constraint could be replaced with some other constraints, such as e.g. requiring each component to have activity 'localized' in time. This problem may be addressed in future work.

# Materials and methods

We will first explain the dPCA algorithm in the most well-behaved case of simultaneously recorded and fully balanced data. A dataset with categorical predictors is called *balanced* when there is the same number of data points for each combination of predictors; in our case this means that there is the same number of trials for each combination of task parameters. This scenario is unlikely in most practical applications where the experimenter often does not have full control over some of the task parameters (such as e.g. animal's decisions). Our suggestion for unbalanced datasets is to use what amounts to a 're-balancing' procedure as explained below. Finally, we will deal with the case of sequentially recorded neurons (all datasets analyzed in this manuscript fall into this category).

## Mathematical notation

In each of the datasets analyzed in this manuscript, trials can be labeled with two parameters: 'stimulus' and 'decision'. Note that a 'reward' label is not needed, because its value can be deduced from the other two due to the deterministic reward protocols in all tasks. In this situation, for each stimulus $s$ (out of $S$) and decision $d$ (out of $Q$), we have a collection of $K$ trials with $N$ neurons recorded in each trial. For each trial $k$ (out of $K$) and neuron $n$ (out of $N$) we have a recorded spike train. We denote the filtered (or binned) spike train by $x(t)$, and assume that it is sampled at $T$ time points $t$. To explicitly denote all task parameters, we will write either $x(t, s, d, k)$ or $x_{tsdk}$ for the filtered spike train of one neuron and $\mathbf{x}_{tsdk}$ for the vector of filtered spike trains of all $N$ neurons. The latter notation is more compact and also highlights the tensorial character of the data.

These data can be thought of as $KSQ$ time-dependent neural trajectories ($K$ trials for each of the $SQ$ conditions) in the $N$-dimensional space $\mathbb{R}^N$ (*Figure 2b*). The number of distinct data points in this $N$-dimensional space is $KSQT$. We collect the full data with all single trials in a matrix $\mathbf{X}$ of size $N \times KSQT$, i.e. $N$ rows and $KSQT$ columns. Averaging all $K$ trials for each neuron, stimulus, and decision, yields mean firing rates (PSTHs) that can be collected in a smaller matrix $\widetilde{\mathbf{X}}$ of size $N \times SQT$.

## Marginalization procedure

Consider one single neuron first. We can decompose its filtered spike trains, $x_{tsdk}$, into a set of averages (which we call *marginalizations*) over various combinations of parameters. We will denote the average over a set of parameters $\{a, b, \ldots\}$ by angular brackets $\langle \cdot \rangle_{ab\ldots}$. Let us define the following marginalized averages:

$$
\begin{aligned}
\bar{x} &= \langle x_{tsdk} \rangle_{tsdk} & &= \bar{x}_{\cdots} \\
\bar{x}_t &= \langle x_{tsdk} - \bar{x} \rangle_{sdk} & &= \bar{x}_{t\cdots} - \bar{x}_{\cdots} \\
\bar{x}_s &= \langle x_{tsdk} - \bar{x} \rangle_{tdk} & &= \bar{x}_{\cdot s\cdot} - \bar{x}_{\cdots} \\
\bar{x}_d &= \langle x_{tsdk} - \bar{x} \rangle_{tsk} & &= \bar{x}_{\cdot\cdot d\cdot} - \bar{x}_{\cdots} \\
\bar{x}_{ts} &= \langle x_{tsdk} - \bar{x} - \bar{x}_t - \bar{x}_s - \bar{x}_d \rangle_{dk} & &= \bar{x}_{ts\cdot} - \bar{x}_{t\cdots} - \bar{x}_{\cdot s\cdot} + \bar{x}_{\cdots} \\
\bar{x}_{td} &= \langle x_{tsdk} - \bar{x} - \bar{x}_t - \bar{x}_s - \bar{x}_d \rangle_{sk} & &= \bar{x}_{t\cdot d\cdot} - \bar{x}_{t\cdots} - \bar{x}_{\cdot\cdot d\cdot} + \bar{x}_{\cdots} \\
\bar{x}_{sd} &= \langle x_{tsdk} - \bar{x} - \bar{x}_t - \bar{x}_s - \bar{x}_d \rangle_{tk} & &= \bar{x}_{\cdot sd\cdot} - \bar{x}_{\cdot s\cdot} - \bar{x}_{\cdot\cdot d\cdot} + \bar{x}_{\cdots} \\
\bar{x}_{tsd} &= \langle x_{tsdk} - \bar{x} - \bar{x}_t - \bar{x}_s - \bar{x}_d - \bar{x}_{ts} - \bar{x}_{td} - \bar{x}_{sd} \rangle_k & &= \bar{x}_{tsd\cdot} - \bar{x}_{ts\cdot} - \bar{x}_{\cdot sd\cdot} - \bar{x}_{t\cdot d\cdot} \\
& & & \quad + \bar{x}_{t\cdots} + \bar{x}_{\cdot s\cdot} + \bar{x}_{\cdot\cdot d\cdot} - \bar{x}_{\cdots} \\
\epsilon_{tsdk} &= x_{tsdk} - \langle x_{tsdk} \rangle_k & &= x_{tsdk} - \bar{x}_{tsd\cdot}.
\end{aligned}
$$

Here $\bar{x}$ is simply the overall mean firing rate of our neuron, $\bar{x}_t$ is the average time-varying firing rate once the overall mean has been subtracted, etc. The right-hand side shows the same averaging procedure in the more explicit form using ANOVA-style notation, in which averages of $x$ over everything apart from the explicitly mentioned parameters, e.g., the stimulus $s$, are denoted by terms of the form $\bar{x}_{\cdot s\cdots}$. One can directly see that the original neural activities are given by the sum of all marginalizations:

$$
x_{tsdk} = \bar{x} + \bar{x}_t + \bar{x}_s + \bar{x}_d + \bar{x}_{ts} + \bar{x}_{td} + \bar{x}_{ds} + \bar{x}_{tsd} + \epsilon_{tsdk}.
$$

This decomposition is identical to the one used in factorial ANOVA (*Rutherford, 2001*; *Christensen, 2011*) where task parameters are called *factors*. The ANOVA literature uses a slightly different notation with task parameters $(t, s, d, k)$ replaced by indices $(i, j, k, l)$ and with Greek letters designating individual terms:

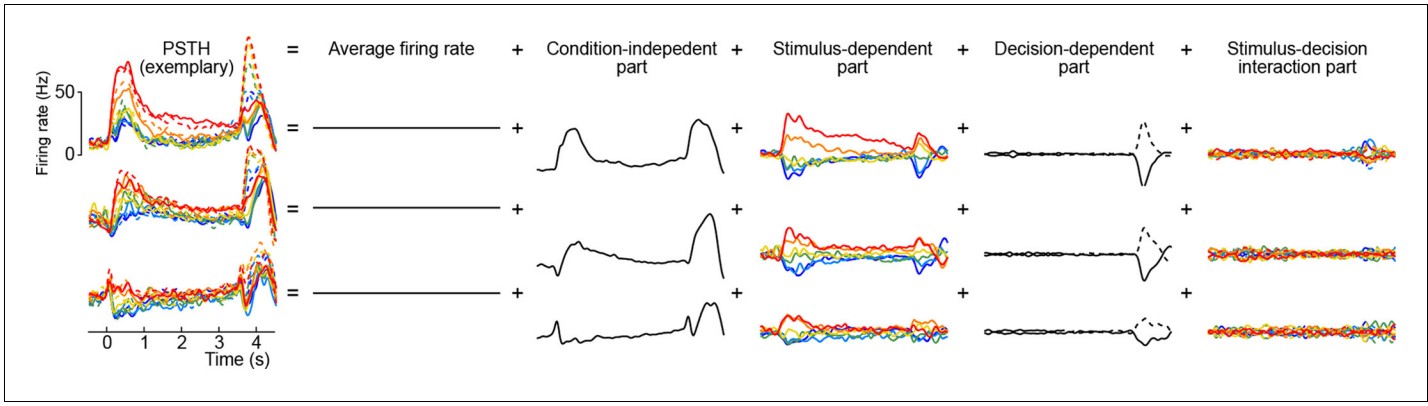

**Figure 8.** Marginalization procedure. PSTHs of three exemplary neurons from the somatosensory working memory task decomposed into marginalizations.

$$x_{ijkl} = \mu + \alpha_i + \beta_j + \gamma_k + \delta_{ij} + \zeta_{jk} + \eta_{ik} + \theta_{ijk} + \epsilon_{ijkl}.$$

We will use our notation, though, to keep the connection with the task parameters more explicit.

For the purposes of demixing neural signals in the context of our datasets, we combine some of these terms together. Indeed, demixing a time-independent pure stimulus term $\bar{x}_s$ from a stimulus-time interaction term $\bar{x}_{ts}$ makes little sense because we expect *all* neural components to change with time. Hence, we group the terms as follows (without changing the notation):

$$x_{tsdk} = \bar{x} + \bar{x}_t + \underbrace{\bar{x}_s + \bar{x}_{ts}}_{\bar{x}_{ts}} + \underbrace{\bar{x}_d + \bar{x}_{td}}_{\bar{x}_{td}} + \underbrace{\bar{x}_{sd} + \bar{x}_{tsd}}_{\bar{x}_{tsd}} + \epsilon_{tsdk}.$$

Here the first term on the right-hand side is the mean firing rate, the last term is the trial-to-trial noise, and we call the other terms condition-independent term, stimulus term, decision term, and stimulus-decision interaction term. This decomposition is illustrated in *Figure 8* for several exemplary neurons (we only show the decomposition of the PSTH part, leaving out the noise term).

We apply this marginalization procedure to every neuron, splitting the whole data matrix $\mathbf{X}$ into parts. Assuming from now on that the data matrix is centered (i.e. $\bar{x} = 0$ for all neurons), we can write the decomposition in the matrix form

$$\mathbf{X} = \mathbf{X}_t + \mathbf{X}_{ts} + \mathbf{X}_{td} + \mathbf{X}_{tsd} + \mathbf{X}_{\text{noise}} = \sum_\phi \mathbf{X}_\phi + \mathbf{X}_{\text{noise}}.$$

Here $t$, $ts$, $td$, and $tsd$ are labels and not indices, and all terms are understood to be matrices of the same $N \times KSQT$ size, so e.g. $\mathbf{X}_t$ is not an $N \times T$ sized matrix, but the full size $N \times KSQT$ matrix with $N \times T$ unique values replicated $KSQ$ times. Crucially, the marginalization procedure ensures that all terms are uncorrelated and that the $N \times N$ covariance matrix $\mathbf{C} = \mathbf{X}\mathbf{X}^\top/(KSQT)$ is linearly decomposed into the sum of covariance matrices from each marginalization (see Appendix A for the proof):

$$\mathbf{C} = \mathbf{C}_t + \mathbf{C}_{ts} + \mathbf{C}_{td} + \mathbf{C}_{tsd} + \mathbf{C}_{\text{noise}} = \sum_\phi \mathbf{C}_\phi + \mathbf{C}_{\text{noise}}.$$

Here all covariance matrices are defined with the same denominator, i.e. $\mathbf{C}_\phi = \mathbf{X}_\phi \mathbf{X}_\phi^\top/(KSQT)$.

## Core dPCA: loss function and algorithm

Given a decomposition $\mathbf{X} = \sum_\phi \mathbf{X}_\phi + \mathbf{X}_{\text{noise}}$, the loss function of dPCA is given by

$$L = \sum_{\phi} L_{\phi}$$

with

$$L_{\phi} = \|\mathbf{X}_{\phi} - \mathbf{F}_{\phi}\mathbf{D}_{\phi}\mathbf{X}\|^2,$$

where each $\mathbf{F}_{\phi}$ is an encoder matrix with $q_{\phi}$ columns and each $\mathbf{D}_{\phi}$ is a decoder matrix with $q_{\phi}$ rows. Here and below, matrix norm signifies Frobenius norm, i.e. $\|\mathbf{X}\|^2 = \sum_i \sum_j X_{ij}^2$. In the remaining discussion, it will often be sufficient to focus on the individual loss functions $L_{\phi}$, in which case we will drop the indices $\phi$ on the decoder and encoder matrices for notational convenience, and simply write $\mathbf{F}$ and $\mathbf{D}$.

Without any additional constraints, the decoder and encoder are only defined up to their product $\mathbf{FD}$ of rank $q$. To make the decomposition unique, we will assume that $\mathbf{F}$ has orthonormal columns and that components are ordered such that their variance (row variance of $\mathbf{DX}$) is decreasing. The reason for this choice will become clear below.

This loss function penalizes the difference between the marginalized data $\mathbf{X}_{\phi}$ and the reconstructed full data $\mathbf{X}$, i.e., the full data projected with the decoders $\mathbf{D}$ onto a low-dimensional latent space and then reconstructed with the encoders $\mathbf{F}$ (see *Video 2*). The loss function thereby favours variance in marginalization $\phi$ and punishes variance coming from all other marginalizations and from trial-to-trial noise. Given that the marginalized averages are uncorrelated with each other, we can make this observation clear by writing,

$$L_{\phi} = \|\mathbf{X}_{\phi} - \mathbf{FDX}\|^2 = \|\mathbf{X}_{\phi} - \mathbf{FDX}_{\phi}\|^2 + \|\mathbf{FD}(\mathbf{X} - \mathbf{X}_{\phi})\|^2.$$

Here the first term corresponds to the non-explained variance in marginalization $\phi$ and the second term corresponds to the variance coming from all other marginalizations and from trial-to-trial noise. The dPCA objective is to minimize both.

We note that the loss function $L_{\phi}$ is of the general form $\|\mathbf{X}_{\phi} - \mathbf{AX}\|^2$, with $\mathbf{A} = \mathbf{FD}$. For an arbitrary $N \times N$ matrix $\mathbf{A}$, minimization of the loss function amounts to a classical regression problem with the well-known ordinary least squares (OLS) solution, $\mathbf{A}_{\mathrm{OLS}} = \mathbf{X}_{\phi}\mathbf{X}^{\top}(\mathbf{XX}^{\top})^{-1}$. In our case, $\mathbf{A} = \mathbf{FD}$ is an $N \times N$ matrix of rank $q$, which we will make explicit by writing $\mathbf{A}_q$. The dPCA loss function therefore amounts to a linear regression problem with an additional rank constraint on the matrix of regression coefficients. This problem is known as *reduced-rank regression (RRR)* (*Izenman, 1975*; *Reinsel and Velu, 1998*; *Izenman, 2008*) and can be solved via the singular value decomposition.

To see this, we write $\mathbf{X}_{\phi} - \mathbf{A}_q\mathbf{X} = (\mathbf{X}_{\phi} - \mathbf{A}_{\mathrm{OLS}}\mathbf{X}) + (\mathbf{A}_{\mathrm{OLS}}\mathbf{X} - \mathbf{A}_q\mathbf{X})$. The first term, $\mathbf{X}_{\phi} - \mathbf{A}_{\mathrm{OLS}}\mathbf{X}$, consists of the regression residuals that cannot be accounted for by any linear transformation of $\mathbf{X}$. It is straightforward to verify that these regression residuals, $\mathbf{X}_{\phi} - \mathbf{A}_{\mathrm{OLS}}\mathbf{X}$, are orthogonal to $\mathbf{X}$ (*Hastie et al., 2009*, Section 3.2) and hence also orthogonal to $(\mathbf{A}_{\mathrm{OLS}} - \mathbf{A}_q)\mathbf{X}$. This orthogonality allows us to split the loss function into two terms,

$$\|\mathbf{X}_{\phi} - \mathbf{A}_q\mathbf{X}\|^2 = \|\mathbf{X}_{\phi} - \mathbf{A}_{\mathrm{OLS}}\mathbf{X}\|^2 + \|\mathbf{A}_{\mathrm{OLS}}\mathbf{X} - \mathbf{A}_q\mathbf{X}\|^2,$$

where the first term captures the (unavoidable) error of the least squares fit while the second term describes the additional loss suffered through the rank constraint. Since the first term does not depend on $\mathbf{A}_q$, the problem reduces to minimizing the second term.

To minimize the second term, we note that the best rank-$q$ approximation to $\mathbf{A}_{\mathrm{OLS}}\mathbf{X}$ is given by its first $q$ principal components (Eckart-Young-Mirsky theorem). Accordingly, if we write $\mathbf{U}_q$ for the matrix of the $q$ leading principal directions (left singular vectors) $\mathbf{u}_i$ of $\mathbf{A}_{\mathrm{OLS}}\mathbf{X}$, then the best approximation is given by $\mathbf{U}_q\mathbf{U}_q^{\top}\mathbf{A}_{\mathrm{OLS}}\mathbf{X}$ and hence $\mathbf{A}_q = \mathbf{U}_q\mathbf{U}_q^{\top}\mathbf{A}_{\mathrm{OLS}}$.

To summarize, the reduced-rank regression problem posed above can be solved in a three-step procedure:

1. Compute the OLS solution $\mathbf{A}_{\mathrm{OLS}} = \mathbf{X}_{\phi}\mathbf{X}^{\top}(\mathbf{XX}^{\top})^{-1}$.

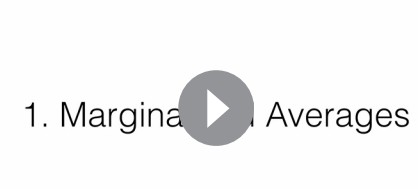

**Video 2.** Illustration of the dPCA algorithm. Illustration of the dPCA algorithm using the somatosensory working memory task.

2. Perform PCA of $\mathbf{A}_{\mathrm{OLS}}\mathbf{X}$ and take the $q$ leading principal components to obtain the best low-rank approximation: $\mathbf{A}_q = \mathbf{U}_q\mathbf{U}_q^\top\mathbf{A}_{\mathrm{OLS}}$ where $\mathbf{U}_q$ is the $N \times q$ matrix of the $q$ leading principal directions (left singular vectors) of $\mathbf{A}_{\mathrm{OLS}}\mathbf{X}$.

3. Factorize the matrix $\mathbf{A}_q$ into decoder and encoder matrices, $\mathbf{A}_q = \mathbf{F}\mathbf{D}$, by choosing $\mathbf{F} = \mathbf{U}_q$ and $\mathbf{D} = \mathbf{U}_q^\top\mathbf{A}_{\mathrm{OLS}}$.

Conveniently, the extracted decoder/encoder pairs do not depend on how many pairs are extracted: the $i$-th pair is given by $\mathbf{f} = \mathbf{u}_i$ and $\mathbf{d} = \mathbf{u}_i^\top\mathbf{A}_{\mathrm{OLS}}$, independent of $q$. Indeed, this feature motivated the above choice that $\mathbf{F}$ should have orthonormal columns.

## Regularization

A standard way to avoid overfitting in regression problems is to add a quadratic penalty to the cost function, which is often called ridge regression (RR). This approach can be used in reduced-rank regression as well. Specifically, we can add a ridge penalty term to the loss function $L_\phi$:

$$L_\phi = \|\mathbf{X}_\phi - \mathbf{F}\mathbf{D}\mathbf{X}\|^2 + \mu\|\mathbf{F}\mathbf{D}\|^2.$$

The RR solution modifies the OLS solution from above to

$$\mathbf{A}_{\mathrm{RR}} = \mathbf{X}_\phi\mathbf{X}^\top\left(\mathbf{X}\mathbf{X}^\top + \mu\mathbf{I}\right)^{-1}.$$

In turn, the reduced-rank solution can be obtained as described above: $\mathbf{F} = \mathbf{U}_q$ and $\mathbf{D} = \mathbf{U}_q^\top\mathbf{A}_{\mathrm{RR}}$ where $\mathbf{U}_q$ are the first $q$ principal directions of $\mathbf{A}_{\mathrm{RR}}\mathbf{X}$.

We found it convenient to define $\mu = (\lambda\|\mathbf{X}\|)^2$, since this makes the values of $\lambda$ comparable across datasets. As explained below, we used cross-validation to select the optimal value of $\lambda$ in each dataset.

## Unbalanced data

The data and variance decomposition carried out by the marginalization procedure can break down when the dataset is unbalanced, i.e., when the number of data points (trials) differs between

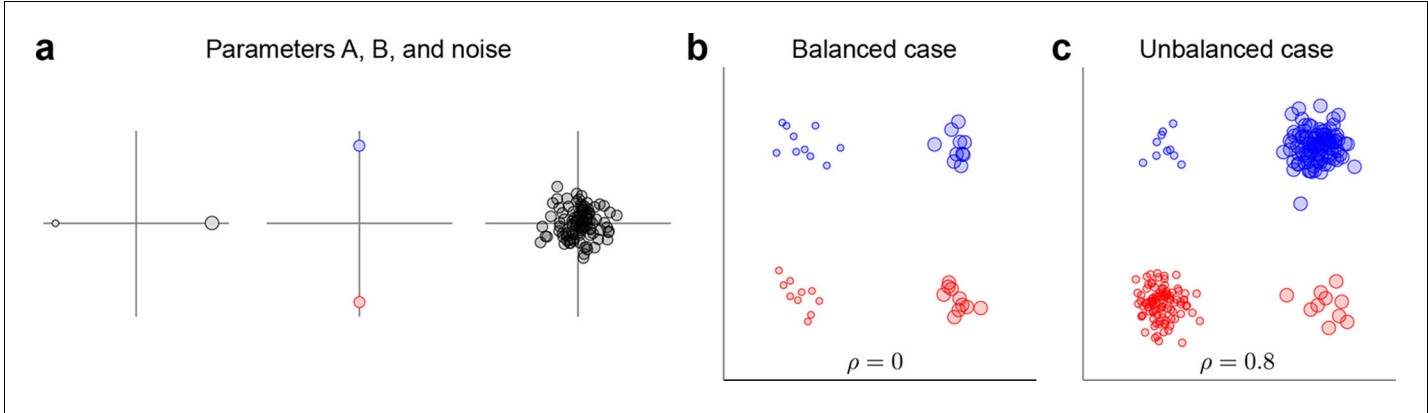

**Figure 9.** Balanced and unbalanced data. (a) In this toy example there are two task parameters (factors), with two possible values each. Parameter A (left) is represented by the size of the dot, parameter B (middle) is represented by the color of the dot, noise is Gaussian with zero mean and zero correlation (right). Interaction term is equal to zero. (b) Balanced case with $N = 10$ data points in each of the four parameter combinations. Overall correlation is zero. (c) Unbalanced case with $N = 10$ for two parameter combinations and $N = 100$ for the other two. Overall correlation is 0.8.

conditions. We illustrate this problem with a two-dimensional toy example in *Figure 9*. We assume two task parameters (factors), each of which can take only two possible values. The overall mean as well as the interaction term are taken to be zero, so that $\mathbf{x}_{ijk} = \mathbf{a}_i + \mathbf{b}_j + \mathbf{e}_{ijk}$. Since the number of trials, $K = K_{ij}$, depends on the condition, the trial index runs through the values $k = 1 \ldots K_{ij}$. As shown in *Figure 9a*, all three terms on the right-hand side exhibit zero correlation between $x_1$ and $x_2$. A balanced dataset with the same number of data points in each of the four possible conditions (*Figure 9b*) also has zero correlation. However, an unbalanced dataset, as shown in *Figure 9c*, exhibits strong positive correlation ($\rho = 0.8$). Accordingly, the covariance matrix of the full data can no longer be split into marginalized covariances. To avoid this and other related problems, we can perform a 're-balancing' procedure by reformulating dPCA in terms of PSTHs and noise covariance.

In the balanced case, the dPCA loss function $L_\phi$ can be rewritten as the sum of two terms with one term depending on the PSTHs and another term depending on the trial-to-trial variations,

$$L_\phi = \|\mathbf{X}_\phi - \mathbf{FDX}\|^2 = \|\mathbf{X}_\phi - \mathbf{FD}(\mathbf{X} - \mathbf{X}_{\mathrm{noise}})\|^2 + \|\mathbf{FDX}_{\mathrm{noise}}\|^2,$$

where we used the fact that $\mathbf{X}_\phi$ and $\mathbf{X} - \mathbf{X}_{\mathrm{noise}}$ are orthogonal to $\mathbf{X}_{\mathrm{noise}}$ (see Appendix A). We now define $\mathbf{X}_{\mathrm{PSTH}} = \mathbf{X} - \mathbf{X}_{\mathrm{noise}}$ which is simply a matrix of the same size as $\mathbf{X}$ with the activity of each trial replaced by the corresponding PSTH. In addition, we observe that the squared norm of any centered data matrix $\mathbf{Y}$ with $n$ data points can be written in terms of its covariance matrix $\mathbf{C}_Y = \mathbf{YY}^\top / n$, namely $\|\mathbf{Y}\|^2 = \mathrm{tr}[\mathbf{YY}^\top] = n\,\mathrm{tr}[\mathbf{C}_Y] = n\,\mathrm{tr}[\mathbf{C}_Y^{1/2}\mathbf{C}_Y^{1/2}] = n\|\mathbf{C}_Y^{1/2}\|^2$, and so

$$L_\phi = \|\mathbf{X}_\phi - \mathbf{FDX}_{\mathrm{PSTH}}\|^2 + KSQT\|\mathbf{FDC}_{\mathrm{noise}}^{1/2}\|^2.$$

The first term consists of $K$ replicated copies: $\mathbf{X}_{\mathrm{PSTH}}$ contains $K$ replicated copies of $\widetilde{\mathbf{X}}$ (which we defined above as the matrix of PSTHs) and $\mathbf{X}_\phi$ contains $K$ replicated copies of $\widetilde{\mathbf{X}}_\phi$ (which we take to be a marginalization of $\widetilde{\mathbf{X}}$, with $\widetilde{\mathbf{X}} = \sum_\phi \widetilde{\mathbf{X}}_\phi$). We can eliminate the replications and drop the factor $K$ to obtain

$$L_\phi = \|\widetilde{\mathbf{X}}_\phi - \mathbf{FD}\widetilde{\mathbf{X}}\|^2 + SQT\|\mathbf{FDC}_{\mathrm{noise}}^{1/2}\|^2.$$

In the unbalanced case, we can directly use this last formulation where all occurrences of $\mathbf{X}$ have been replaced by $\widetilde{\mathbf{X}}$. This is especially useful for neural data, where some combinations of task parameters may occur more often than others. The 're-balanced' dPCA loss function treats all parameter combinations as equally important, independent of their occurrence frequency. It stands to reason to 're-balance' the noise covariance matrix as well by defining it as follows:

$$\widetilde{\mathbf{C}}_{\mathrm{noise}} = \frac{1}{SQT}\sum_{sdt}\mathbf{C}_{\mathrm{noise}}(s,d,t) = \big\langle\mathbf{C}_{\mathrm{noise}}(s,d,t)\big\rangle_{sdt},$$

where $\mathbf{C}_{\mathrm{noise}}(s,d,t)$ is the covariance matrix for the $(s,d,t)$ parameter combination. This formulation, again, treats noise covariance matrices from different parameter combinations as equally important, independent of how many data points there are for each parameter combination.

Putting everything together and including the regularization term as well, we arrive at the following form of the dPCA loss function:

$$L_\phi = \|\widetilde{\mathbf{X}}_\phi - \mathbf{FD}\widetilde{\mathbf{X}}\|^2 + SQT\|\mathbf{FD}\widetilde{\mathbf{C}}_{\mathrm{noise}}^{1/2}\|^2 + \mu\|\mathbf{FD}\|^2.$$

This loss function can be minimized as described in the previous section. Specifically, the full rank solution with $\mathbf{A} = \mathbf{FD}$ becomes

$$\mathbf{A}_{\mathrm{RR}} = \widetilde{\mathbf{X}}_\phi\widetilde{\mathbf{X}}^\top\big(\widetilde{\mathbf{X}}\widetilde{\mathbf{X}}^\top + SQT\cdot\widetilde{\mathbf{C}}_{\mathrm{noise}} + \mu\mathbf{I}\big)^{-1}.$$

The reduced-rank solution can then be obtained by setting $\mathbf{F} = \mathbf{U}_q$ and $\mathbf{D} = \mathbf{U}_q^\top\mathbf{A}$, where $\mathbf{U}_q$ are the first $q$ principal directions of $\mathbf{A}_{\mathrm{RR}}\widetilde{\mathbf{X}}$.

## Missing data

Even when using the re-balanced formulation of the loss function, we still need data from all possible parameter combinations. In neurophysiological experiments, however, one may run into situations where not all combinations of stimuli could be presented to an animal before it decided to abort the task, or where an animal never carried out a particular decision, etc. This problem is particularly severe if individual task parameters can take many values. What should one do in these cases? The key problem here is that dPCA as formulated above makes no assumptions about how the firing rates of individual neurons depend on the task parameters. (Nor is there an explicit assumption about how the demixed components depend on the task parameters.) If some task conditions have not been recorded, then the only way out is to add more assumptions, or, more formally, to replace the non-parametric estimates of individual neural firing rates (or demixed components) by parametric estimates. We could for instance fit a simple linear model to the firing rate of each neuron at each time step (*Mante et al., 2013*; *Brody et al., 2003*),

$$x(t,s,d) = \alpha(t) + \beta(t)s + \gamma(t)d + \epsilon$$

and then use this model to 'fill in' the missing data. More sophisticated ways of dealing with missing data could be envisaged as well and may provide a venue for future research.

## Sequentially recorded data

For sequentially recorded datasets, the matrix $\mathbf{X}$ cannot be meaningfully constructed. However, we can still work with the PSTH matrix $\widetilde{\mathbf{X}}$ that can be decomposed into marginalizations: $\widetilde{\mathbf{X}} = \sum_\phi \widetilde{\mathbf{X}}_\phi$. Consequently, we can use the same formulation of the loss function as in the simultaneously recorded unbalanced case (see above). The only difference is that the noise covariance matrix is not available (noise correlations cannot be estimated when neurons are recorded in different sessions). In this manuscript we took as $\mathbf{C}_{\mathrm{noise}}$ the diagonal matrix with individual noise variances of each neuron on the diagonal. We used the re-balanced version $\widetilde{\mathbf{C}}_{\mathrm{noise}}$ (average noise covariance matrix across all conditions), but found that the difference between re-balanced and non-rebalanced noise covariance matrices was always minor and did not noticeably influence the dPCA solutions.

## Variance calculations

As all datasets analyzed in this manuscript were sequentially recorded, we always reported fractions of the PSTH variance (as opposed to the total PSTH+noise variance) explained by our components, i.e. fractions of variance explained in $\widetilde{\mathbf{X}}$. We defined the fraction of explained variance in a standard way:

$$R^2 = \frac{\|\widetilde{\mathbf{X}}\|^2 - \|\widetilde{\mathbf{X}} - \mathbf{F}\mathbf{D}\widetilde{\mathbf{X}}\|^2}{\|\widetilde{\mathbf{X}}\|^2}.$$

This formula can be used to compute the fraction of variance explained by each dPCA component (by plugging in its encoder $\mathbf{f}$ and decoder $\mathbf{d}$); these are the numbers reported on *Figures 3–6b,d* and used to order the components. The same formula can be used to compute the cumulative fraction of variance explained by the first $q$ components (by stacking their encoders and decoders as columns and rows of $\mathbf{F}$ and $\mathbf{D}$ respectively); these are the numbers reported on *Figures 3–6c*. Note that the cumulative explained variance is close to the sum of individually explained variances but not exactly equal to it since the dPCA components are not completely uncorrelated. The same formula holds for standard PCA using $\mathbf{F} = \mathbf{D}^\top = \mathbf{U}_{\mathrm{pca}}$, i.e., the matrix of stacked together principal directions (*Figures 3–6c*).

Using the decomposition $\widetilde{\mathbf{X}} = \sum_\phi \widetilde{\mathbf{X}}_\phi$, we can split the fraction of explained variance into additive contributions from different marginalizations:

$$R^2 = \sum_\phi \frac{\|\widetilde{\mathbf{X}}_\phi\|^2 - \|\widetilde{\mathbf{X}}_\phi - \mathbf{F}\mathbf{D}\widetilde{\mathbf{X}}_\phi\|^2}{\|\widetilde{\mathbf{X}}\|^2}.$$

We used this decomposition to produce the bar plots in *Figures 3–6d*, showing how the explained variance of each single dPCA component is split between marginalizations.

Following the approach of *Machens et al. (2010)*, we note that our PSTH estimates $\widetilde{\mathbf{X}}$ must differ from the 'true' underlying PSTHs due to the finite amount of recorded trials. Hence, some fraction of the total variance of $\widetilde{\mathbf{X}}$ is coming from this residual noise. We can estimate this fraction as follows. Our estimate of the noise variance of the $n$-th neuron is given by $\widetilde{C}_{nn}$, the $n$-th diagonal element of $\widetilde{\mathbf{C}}_{\mathrm{noise}}$. There are on average $\widetilde{K}_n = \frac{1}{SQ}\sum K_{nsd}$ trials being averaged to compute the PSTHs for this neuron. So a reasonable estimate of the residual noise variance of the $n$-th neuron is $\widetilde{C}_{nn}/\widetilde{K}_n$. Accordingly, we define the total residual noise sum of squares as

$$\Theta = SQT \cdot \sum_n \frac{\widetilde{C}_{nn}}{\widetilde{K}_n}.$$

In turn, the fraction of total *signal variance* is computed as $1 - \Theta/\|\widetilde{\mathbf{X}}\|^2$ which is the dashed line shown in *Figures 3–6c*. Note that each component likewise has contributions from both signal and noise variance, and hence the fraction of total signal variance does not constitute an upper bound on the number of components.

The residual noise variance is not split uniformly across marginalizations: the fraction falling into marginalization $\phi$ is proportional to the respective number of degrees of freedom, $K_\phi$. This can be explicitly computed; for a centered dataset with $S$ stimuli, $Q$ decisions, and $T$ time points the total number of degrees of freedom (per neuron) is $SQT - 1$ and is split into $T - 1$ for time, $ST - T$ for stimulus, $QT - T$ for decision, and $SQT - ST - QT + T$ for the stimulus-decision interaction (compare with the formulas in the Marginalization Procedure section). Accordingly, we computed the residual noise sum of squares falling into marginalization $\phi$ as

$$\Theta_\phi = \frac{K_\phi}{SQT - 1}\Theta.$$

The pie charts in *Figures 3–6d* show the amount of variance in each marginalization, with estimated contributions of the residual noise variance subtracted: $\left(\|\widetilde{\mathbf{X}}_\phi\|^2 - \Theta_\phi\right) / \left(\|\widetilde{\mathbf{X}}\|^2 - \Theta\right)$. To display the percentage values on the pie charts, percentages were rounded using the 'largest remainder method', so that the sum of the rounded values remained 100%.

### Demixing indices

We defined the *demixing index* of each component as $\max_\phi\{\|\mathbf{d}\widetilde{\mathbf{X}}_\phi\|^2\}/\|\mathbf{d}\widetilde{\mathbf{X}}\|^2$. This index can range from 1/4 to 1 (since there are four marginalizations) and the closer it is to 1, the better demixed the component is. As an example, for the somatosensory working memory dataset, the average demixing index over the first 15 PCA components is 0.76±0.16 (mean±SD), and over the first 15 dPCA components is 0.98±0.02, which means that dPCA achieves much better demixing ($p = 0.0002$, Mann-Whitney-Wilcoxon ranksum test). For the first 15 components of dPCA-2011 (*Brendel et al., 2011*) it was 0.95±0.03, significantly less than for the current dPCA ($p = 0.0008$). This difference may seem small, but is clearly visible in the projections by the naked eye. For comparison, the average demixing index of individual neurons in this dataset is 0.55±0.18. In other datasets these numbers are similar, and the same differences were significant in all cases.

### Angles between dPCs

In *Figures 3–6e*, stars mark the pairs of components whose encoding axes $\mathbf{f}_1$ and $\mathbf{f}_2$ are significantly and robustly non-orthogonal. These were identified as follows: In Euclidean space of $N$ dimensions, two random unit vectors (from a uniform distribution on the unit sphere) have dot product (cosine of the angle between them) distributed with mean zero and standard deviation $N^{-1/2}$. For large $N$ the distribution is approximately Gaussian. To avoid the problems inherent to multiple comparisons, we chose a conservative significance level of $p<0.001$, which means that two axes are significantly non-orthogonal if $|\mathbf{f}_1 \cdot \mathbf{f}_2| > 3.3/N^{1/2}$.

Coordinates of $\mathbf{f}_1$ quantify how much this component contributes to the activity of each neuron. Hence, if cells exhibiting one component also tend to exhibit another, the dot product between the axes $\mathbf{f}_1 \cdot \mathbf{f}_2 > 0$ is positive (note that $\mathbf{f}_1 \cdot \mathbf{f}_2$ is approximately equal to the correlation between the coordinates of $\mathbf{f}_1$ and $\mathbf{f}_2$). Sometimes, however, the dot product has large absolute value only due to

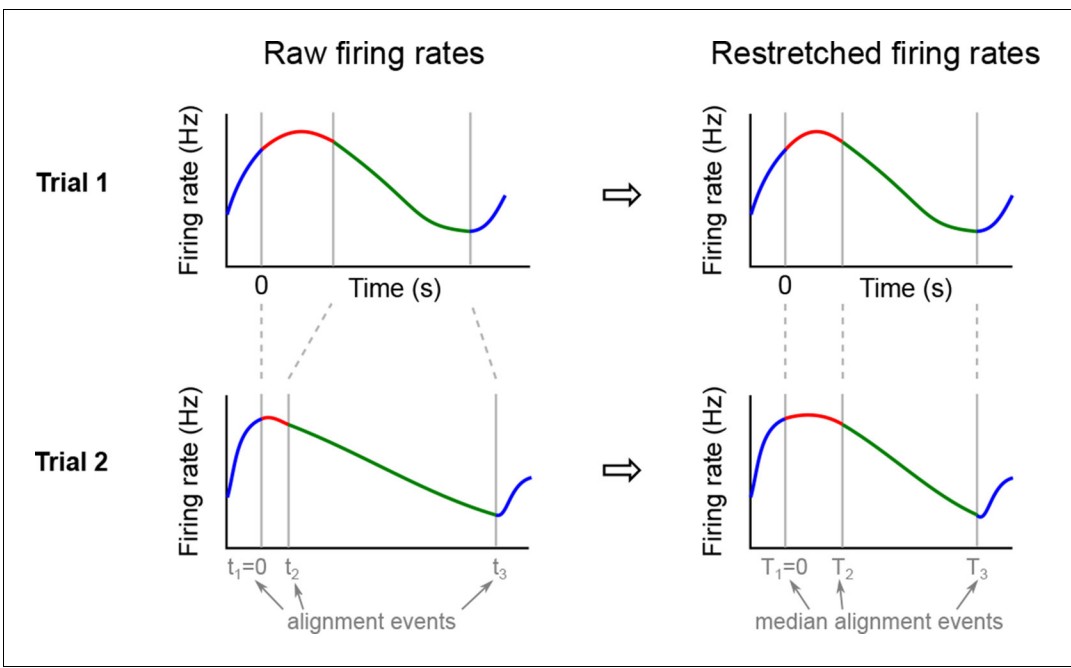

**Figure 10.** Re-stretching (time warping) procedure. We defined several alignment events (such as odour poke in, odour poke out, etc.) and for each trial found the times $t_i$ of these events. After aligning all trials on $t_1 = 0$ (left) we computed median times $T_i$ for all other events. Then for each trial we re-stretched the firing rate on each interval $[t_i, t_{i+1}]$ to align it with $[T_i, T_{i+1}]$ (right). After such re-stretching, all events are aligned and the trials corresponding to one condition can be averaged.

several outlying cells. To ease interpretation, we marked with stars only those pairs of axes for which the Kendall (robust) correlation was significant at $p < 0.001$ level (in addition to the above criterion on $\mathbf{f}_1 \cdot \mathbf{f}_2$).

## Experimental data

Brief descriptions of experimental paradigms are provided in the Results section and readers are referred to the original publications for all further details. Here we describe the selection of animals, sessions, and trials for the present manuscript. In all experiments neural recordings were obtained in multiple sessions, so most of the neurons were not recorded simultaneously. All four datasets used in this manuscript have been made available at http://crcns.org (*Romo et al., 2016*; *Constantinidis et al., 2016*; *Feierstein et al., 2016*; *Uchida et al., 2016*).

1. Somatosensory working memory task in monkeys (*Romo et al., 1999*; *Brody et al., 2003*). We used data from two monkeys (code names RR14 and RR15) that were trained with the same set of vibratory frequencies, and we selected only the sessions where all six frequencies {10, 14, 18, 26, 30, 34} Hz were used for the first stimulation (other sessions were excluded). Monkeys made few mistakes (overall error rate was 6%), and here we analyzed only correct trials. Monkey RR15 had an additional 3 s delay after the end of the second stimulation before it was cued to provide the response. Using the data from monkey RR13 (that experienced a different frequency set) led to very similar dPCA components (data not shown).
2. Visuospatial working memory task in monkeys (*Qi et al., 2011*; *Meyer et al., 2011*; *Qi et al., 2012*). We used the data from two monkeys (code names AD and EL) that were trained with the same spatial task. Monkeys made few mistakes (overall error rate was 8%), and here we analysed only correct trials. The first visual stimulus was presented at 9 possible spatial locations arranged in a 3×3 grid (*Figure 4a*); here we excluded all the trials where the first stimulus was presented in the centre position.
3. Olfactory discrimination task in rats (*Feierstein et al., 2006*). We used the data from all five rats (code names N1, P9, P5, T5, and W1). Some rats were trained with two distinct odors, some with four, some with six, and one rat experienced mixtures of two fixed odors in varying

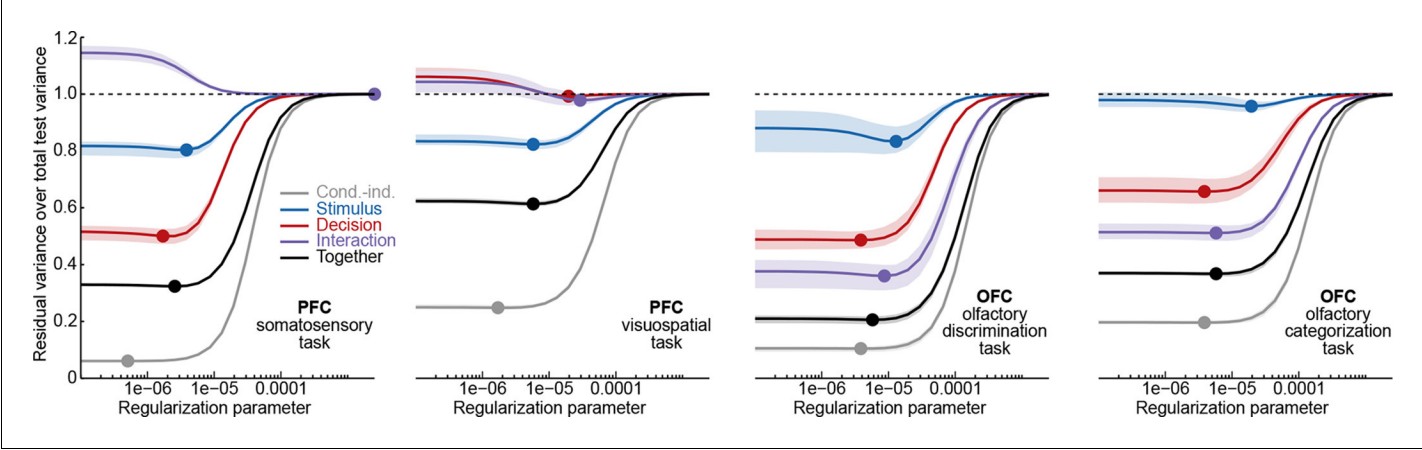

**Figure 11.** Cross-validation errors depending on the regularization parameter $\lambda$. Each subplot corresponds to one dataset and shows mean (solid lines) and min/max (boundaries of shaded regions) of the relative cross-validation errors for ten repetitions. Different colors refer to different marginalizations (see legend), the minima are marked by dots. Black color shows all marginalizations together, i.e. $L_{CV}(\lambda)$.

proportions. In all cases each odor was uniquely associated with one of the two available water ports (left/right). Following the original analysis (*Feierstein et al., 2006*), we grouped all odors associated with the left/right reward together as a 'left/right odor'. For most rats, caproic acid and 1-hexanol (shown in *Figures 5–6a*) were used as the left/right odor. We excluded from the analysis all trials that were aborted by rats before reward delivery (or before waiting 0.2 s at the reward port for the error trials).

4. Olfactory categorization task in rats (*Kepecs et al., 2008*). We used the data from all three rats (code names N1, N48, and N49). Note that recordings from one of the rats (N1) were included in both this and previous datasets; when we excluded it from either of the datasets, the results stayed qualitatively the same (data not shown). We excluded from the analysis all trials that were aborted by rats before reward delivery (or before waiting 0.3 s at the reward port for the error trials).

## Selection of neurons

For our analysis, we only selected neurons which had been recorded in each possible condition (combination of parameters), which avoids the missing data problems explained above. Additionally, we required that in each condition there were at least $K_{min} > 1$ trials, to reduce the standard error of the mean when averaging over trials, and also for cross-validation purposes. The cutoff was set to $K_{min} = 5$ for both working memory datasets, and to $K_{min} = 2$ for both olfactory datasets (due to less neurons available).

We have further excluded very few neurons with mean firing rates over 50 Hz, as such neurons can bias the variance-based analysis. Firing rates above 50 Hz were atypical in all datasets (number of excluded neurons for each dataset: 5 / 2 / 1 / 0). This exclusion had a minor positive effect on the components. We did not apply any variance-stabilizing transformations, but if the square-root transformation was applied, the results stayed qualitatively the same (data not shown).

No other pre-selection of neurons was used. This procedure left 832 neurons (230 / 602 for individual animals, order as above) in the somatosensory working memory dataset, 956 neurons (182 / 774) in the visuospatial working memory dataset, 437 neurons in the olfactory discrimination dataset (166 / 30 / 9 / 106 / 126), and 214 neurons in the olfactory categorization dataset (67 / 38 / 109).

## Preprocessing of the neural data

The spike trains were filtered with a Gaussian kernel ($\sigma = 50$ ms) and sampled at 100 Hz to produce single-trial instantaneous firing rates.

In the visuospatial working memory dataset we identified the preferred location of each neuron as the location that evoked maximum mean firing rate in the 500 ms time period while the first stimulus was shown. The neural tuning was shown before to have a symmetric bell shape (*Qi et al.,*

*2011*; *Meyer et al., 2011*), with each neuron having its own preferred location. We then re-sorted the trials (separately for each neuron) such that only five distinct stimuli were left: preferred location, 45°, 90°, 135°, and 180° away from the preferred location.

In both olfactory datasets trials were self-paced. Accordingly, trials last different amounts of time, and firing rates cannot simply be averaged over trials. We used the following time warping (re-stretching) procedure to equalize the length of all trials and to align several events of interest (*Figure 10*) separately in each dataset. We defined five alignment events: odor poke in, odor poke out, water poke in, reward delivery, and water poke out. First, we aligned all trials on odor poke in ($T_1 = 0$) and computed median times of the four other events $T_i$, $i = 2 \ldots 5$ (for the time of reward delivery, we took the median over all correct trials). Second, we set $\Delta T$ to be the minimal waiting time between water port entry and reward delivery across the whole experiment ($\Delta T = 0.2$ s for the olfactory discrimination task and $\Delta T = 0.3$ s for the olfactory categorization task). Finally, for each trial with instantaneous firing rate $x(t)$ we set $t_i$, $i = 1 \ldots 5$, to be the times of alignment events on this particular trial (for error trials we took $t_4 = t_3 + \Delta T$), and stretched $x(t)$ along the time axis in a piecewise-linear manner to align each $t_i$ with the corresponding $T_i$.

We made sure that time warping did not introduce any artifacts by considering an alternative procedure, where short ($\pm 450$ ms) time intervals around each $t_i$ were cut out of each trial and concatenated together; this procedure is similar to the pooling of neural data performed in the original studies (*Feierstein et al., 2006*; *Kepecs et al., 2008*). The dPCA analysis revealed qualitatively similar components (data not shown).

## Cross-validation to select regularization parameter

As noted above, we renormalized the regularization parameter $\mu = (\lambda \|\mathbf{X}\|)^2$, and then used cross-validation to find the optimal value of $\lambda$ for each dataset. To separate the data into training and testing sets, we held out one random trial for each neuron in each condition as a set of $SQ$ test 'pseudo-trials' $\mathbf{X}_{\text{test}}$ (as the neurons were not recorded simultaneously, we do not have recordings of all $N$ neurons in any actual trial). Remaining trials were averaged to form a training set of PSTHs $\widetilde{\mathbf{X}}_{\text{train}}$ and an estimate of the noise covariance matrix $\widetilde{\mathbf{C}}_{\text{train}}$. Note that $\mathbf{X}_{\text{test}}$ and $\widetilde{\mathbf{X}}_{\text{train}}$ have the same dimensions.

We then performed dPCA on $\widetilde{\mathbf{X}}_{\text{train}}$ for various values of $\lambda$ between $10^{-7}$ and $10^{-3}$ (on a logarithmic grid). For each $\lambda$, we selected ten components in each marginalization (i.e. 40 components in total) to obtain $\mathbf{F}_\phi(\lambda)$ and $\mathbf{D}_\phi(\lambda)$, and computed the normalized reconstruction error $L_{\text{CV}}(\lambda)$ on the test set (see below). We repeated this procedure ten times for different train-test splittings and averaged the resulting functions $L_{\text{CV}}(\lambda)$. In all cases the average function $\bar{L}_{\text{CV}}(\lambda)$ had a clear minimum (*Figure 11*) that we selected as the optimal $\lambda$. The values of $\lambda$ selected for each dataset were $2.6 \cdot 10^{-6}$ / $5.8 \cdot 10^{-6}$ / $5.8 \cdot 10^{-6}$ / $5.8 \cdot 10^{-6}$. We also performed the same procedure in each marginalization separately, but in all datasets the optimal values of $\lambda$ were similar across marginalizations (*Figure 11*). We therefore chose to use the same value of $\lambda$ for all marginalizations.

Interestingly, for all our datasets $\min_\lambda\{\bar{L}_{\text{CV}}(\lambda)\}$ was only slightly smaller than $\bar{L}_{\text{CV}}(0)$, so the regularization term had almost no influence. Presumably, this result stems from our diagonal (and thus non-singular) noise covariance matrices, and therefore does not necessarily hold for simultaneously recorded data.

To compute $L_{\text{CV}}(\lambda)$, we used $\mathbf{X}_{\text{test}}$ to predict $\widetilde{\mathbf{X}}_{\text{train}}$:

$$L_{\text{CV}}(\lambda) = \frac{\sum_\phi \|\widetilde{\mathbf{X}}_{\text{train},\phi} - \mathbf{F}_\phi(\lambda)\mathbf{D}_\phi(\lambda)\mathbf{X}_{\text{test}}\|^2}{\|\widetilde{\mathbf{X}}_{\text{train}}\|^2}.$$

This is the residual training-set variance not explained by the test data. Note that it would not make sense to exchange $\mathbf{X}_{\text{test}}$ and $\widetilde{\mathbf{X}}_{\text{train}}$ in this formula: the decoder and encoder are fitted to the training data, and should only be applied to the test data for the purposes of cross-validation. An alternative approach, in which we predicted the test data rather than the training data, yielded similar results (data not shown).

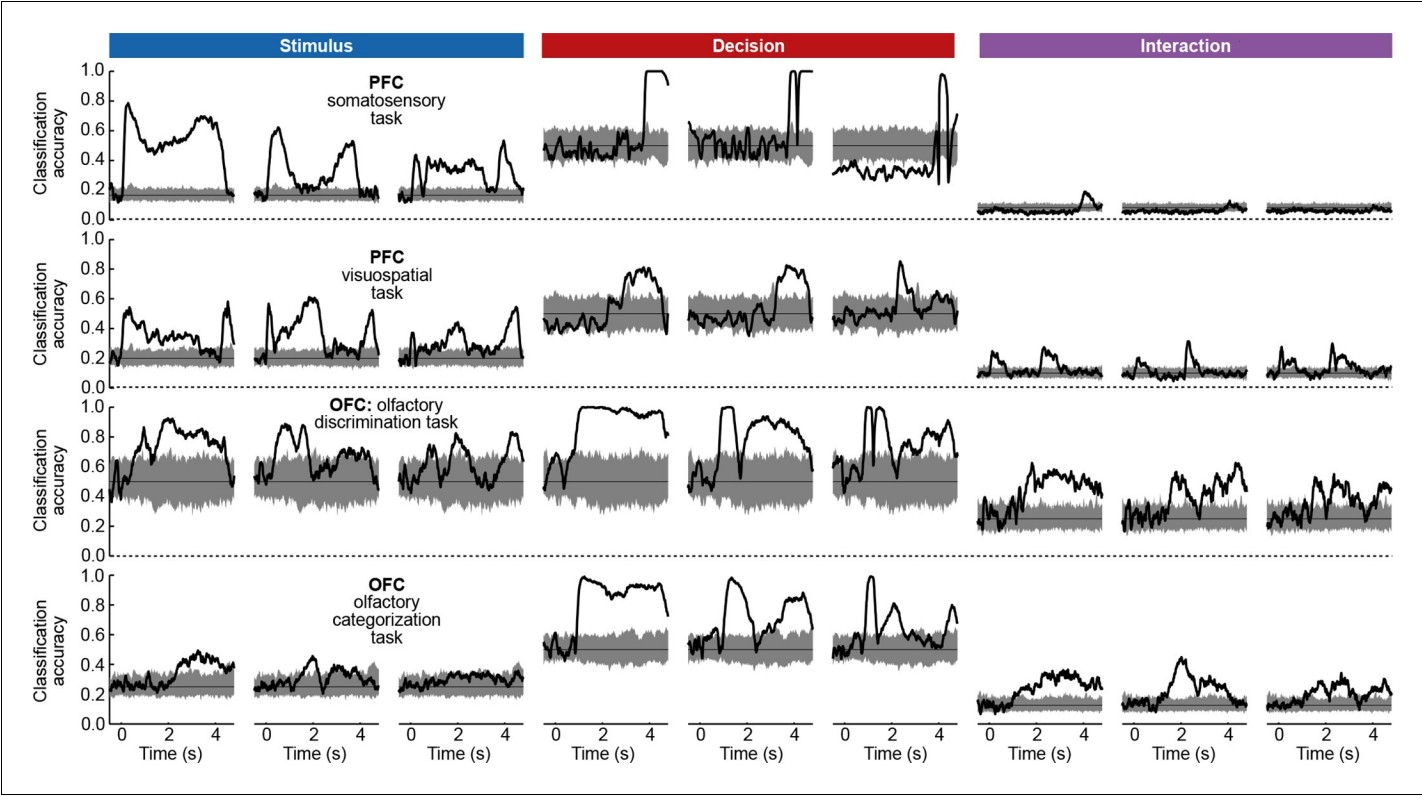

**Figure 12.** Cross-validated time-dependent classification accuracies of linear classifiers (black lines) given by the first three stimulus/decision/interaction dPCs (columns) in each dataset (rows). Shaded gray regions show distribution of classification accuracies expected by chance as estimated by 100 iterations of shuffling procedure.

## Cross-validation to measure classification accuracy

We used decoding axis $\mathbf{d}$ of each dPC in stimulus, decision, and interaction marginalizations as a linear classifier to decode stimulus, decision, or condition respectively. Black lines on *Figures 3–6b* show time periods of significant classification. A more detailed description follows below.

We used 100 iterations of stratified Monte Carlo leave-group-out cross-validation, where on each iteration we held out one trial for each neuron in each condition as a set of $SQ$ test 'pseudo-trials' $\mathbf{X}_{\text{test}}$ and averaged over remaining trials to form a training set $\widetilde{\mathbf{X}}_{\text{train}}$ (see above). After running dPCA on $\widetilde{\mathbf{X}}_{\text{train}}$, we used decoding axes of the first three stimulus/decision/interaction dPCs as a linear classifier to decode stimulus/decision/condition respectively. Consider e.g. the first stimulus dPC: first, for each stimulus, we computed the mean value of this dPC separately for every time-point. Then we projected each test trial on the corresponding decoding axis and classified it at each time-point according to the closest class mean. The proportion of test trials (out of $SQ$) classified correctly resulted in a time-dependent classification accuracy, which we averaged over 100 cross-validation iterations. Note that this is a stratified procedure: even though in reality some conditions have many fewer trials than others, here we classify exactly the same number of 'pseudo-trials' per condition. At the same time, as the coordinates of individual data points in each pseudo-trial are pooled from different sessions, the influence of noise correlations on the classification accuracies is neglected, similar to *Meyers et al. (2012)*.

We then used 100 shuffles to compute the distribution of classification accuracies expected by chance. On each iteration and for each neuron, we shuffled all available trials between conditions, respecting the number of trials per condition (i.e. all $\sum_{sd} K_{nsd}$ trials were shuffled and then randomly assigned to the conditions such that all values $K_{nsd}$ stayed the same). Then exactly the same classification procedure as above (with 100 cross-validation iterations) was applied to the shuffled dataset

to find mean classification accuracy for the first stimulus, decision, and interaction components. All 100 shuffling iterations resulted in a set of 100 time-dependent accuracies expected by chance.

The time periods when actual classification accuracy exceeded all 100 shuffled decoding accuracies in at least ten consecutive time bins are marked by black lines on *Figures 3–6*. Components without any periods of significant classification are not shown. See *Figure 12* for classification accuracies in each dataset. The Monte Carlo computations took ~8 hr for each of the larger datasets on a 6 core 3.2 Ghz Intel i7-3930K processor.

## Implementation of classical approaches (*Figure 1*)

The two-way ANOVA shown in *Figure 1c–e* was performed as follows. The two factors were stimulus (with six levels) and decision (with two levels), the interaction term was included, and a separate ANOVA was run for the firing rate of each neuron at each time point. Significance level was set at $\alpha = 0.05$. Effect size was defined as partial omega squared with a sign given by the sign of the correlation coefficient between firing rate and the corresponding parameter. It can take values between $-1$ and 1, with 0 meaning no effect. For one-way ANOVA with a single two-level factor (which is a t-test), it would reduce to the signed $R^2$ between firing rate and factor level.

For *Figure 1f–g*, we ran linear regressions for the firing rate of each neuron at $t = 0.25$ s and $t = 3.75$ s, taking F1 stimulus value in Hz as one predictor, decision as another one, and including an interaction effect. Predictors were standardized (so regression coefficients in *Figure 1g* are standardized coefficients). The components shown in *Figure 1f* were constructed following the 'targeted dimensionality reduction' method presented in *Mante et al. (2013)* (see below for more details).

To compute the proportion of explained PSTH variance, we arranged the two components (obtained by either method) into a matrix $\mathbf{Z}$ of $2 \times SQT$ size. Both the PSTH data matrix $\widetilde{\mathbf{X}}$ and $\mathbf{Z}$ were centered by subtracting row means. Linear regression was used to find reconstruction weights $\mathbf{B} = \widetilde{\mathbf{X}}\mathbf{Z}^\top(\mathbf{Z}\mathbf{Z}^\top)^{-1}$ minimizing reconstruction error $\|\widetilde{\mathbf{X}} - \mathbf{B}\mathbf{Z}\|^2$. Then the proportion of explained variance was computed as $R^2 = 1 - \|\widetilde{\mathbf{X}} - \mathbf{B}\mathbf{Z}\|^2/\|\widetilde{\mathbf{X}}\|^2$.

The PCA on *Figure 1i–k* was done on the centered PSTH data matrix $\widetilde{\mathbf{X}}$. Let its singular value decomposition be $\widetilde{\mathbf{X}} = \mathbf{U}\mathbf{S}\mathbf{V}^\top$. Then each subplot on *Figure 1j* is a histogram of elements of one column of $\mathbf{U}$ and each subplot on *Figure 1j* is one column of $\mathbf{V}$.

## Comparison of dPCA to PCA in each marginalization

To understand the differences of dPCA with respect to other (demixing) methods, we will make several explicit comparisons. The first method we will consider is performing a series of standard PCAs in each marginalization separately. This procedure can be understood in two ways: after performing PCA on $\mathbf{X}_\phi$ and obtaining the matrix $\mathbf{U}_\phi$ for the $k$ leading principal directions, we can use this matrix to project either the marginalized data or the full data.

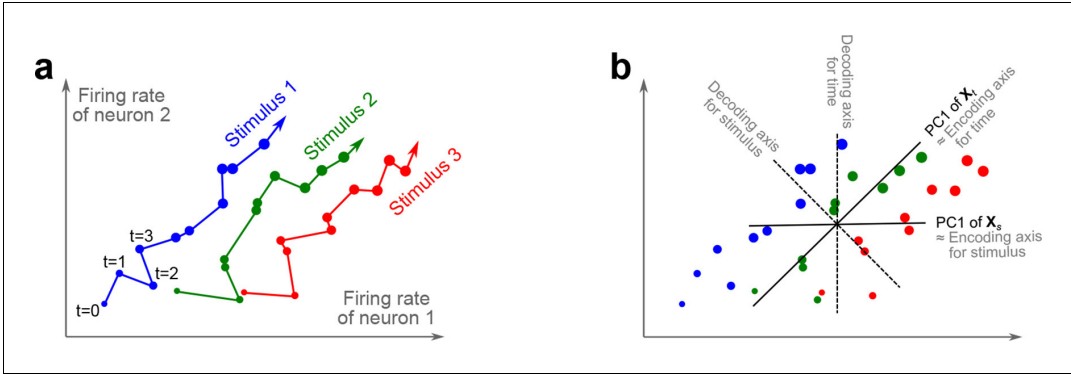

**Figure 13.** Toy example illustrating the pseudo-inverse intuition. (a) Firing rate trajectories of two neurons for three different stimuli. (b) Same data with dPCA decoding and encoding axes. The encoding axes are approximately equivalent to the axes of the principal components in this case.

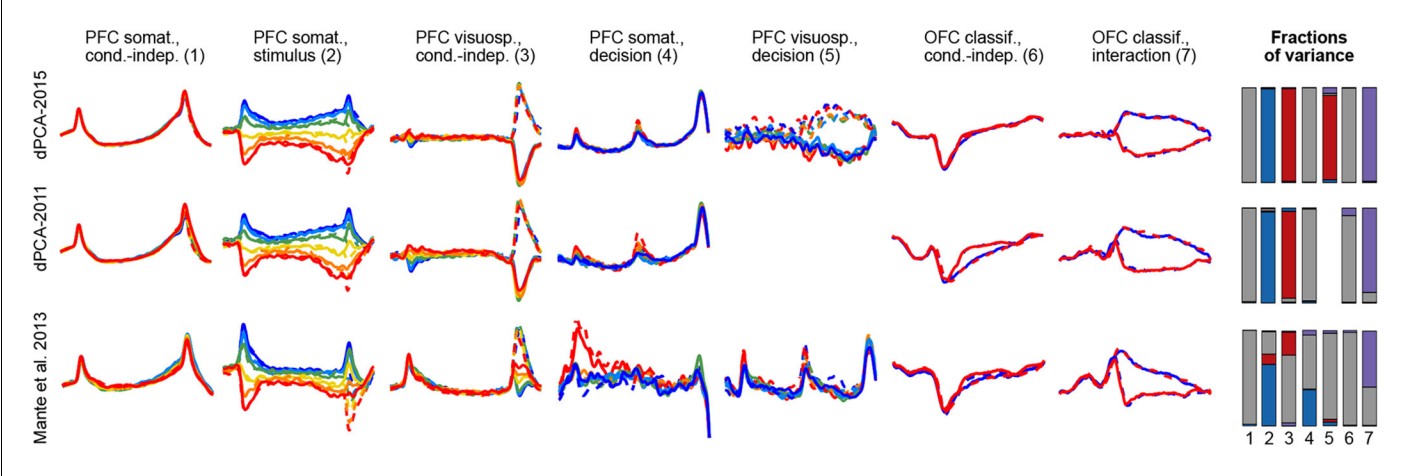

**Figure 14.** Some demixed components as given by three different demixing methods (rows) in various datasets and marginalizations (columns). Empty subplots mean that the corresponding method did not find any components. All projections were $z$-scored to make them of the same scale. Barplots on the right show fractions of variance in each marginalization for each component (stimulus in blue, decision in red, interaction in purple, condition-independent in gray): $\|\mathbf{d}\widetilde{\mathbf{X}}_{\phi}\|^2 / \|\mathbf{d}\widetilde{\mathbf{X}}\|^2$. Barplots consisting of a single colour correspond to perfect demixing.

In the first case we obtain the principal components of the corresponding marginalization, $\mathbf{U}_{\phi}\mathbf{X}_{\phi}$. However, while these components provide a particular decomposition or visualization of the data, they do not constitute *readouts* of the neural activity, since they are based on projecting the marginalized data. One particular advantage of the dPCA formulation is that it operates on the raw data, so that the decoders (and encoders) can actually be used on single trials. In turn, the visualization of the data found through dPCA also provides insights into the utility of the respective population code for the brain.

In the second case we obtain $\mathbf{U}_{\phi}\mathbf{X}$ components from the full data, so that these components could be obtained by projecting single-trial activities. However, now there is no guarantee that these components will be demixed. For a simple counter-example, consider *Figure 13*: the stimulus marginalization $\mathbf{X}_s$ consists of three points (one for each stimulus) located roughly on a horizontal axis, and so the first principal axis of $\mathbf{X}_s$ is roughly horizontal. It is easy to see that the projection of the full data onto this axis will be not only stimulus-, but also time-dependent.

Nonetheless, we can obtain a reasonable approximation to the dPCA solution using PCA in each marginalization. Namely, $\mathbf{U}_{\phi}$ can be taken to constitute the encoders $\mathbf{F}_{\phi}$. In turn, the decoders $\mathbf{D}_{\phi}$ are obtained by a pseudo-inverse $\mathbf{D} = \mathbf{U}^{+}$, where $\mathbf{U}$ is a matrix with $4k$ columns obtained by joining together all $\mathbf{U}_{\phi}$. We found that this procedure provides a close approximation of the actual decoder and encoder matrices, provided one chooses a reasonable value of $k$: choosing $k$ too small results in poor demixing, and choosing $k$ too large results in overfitting. In our datasets, $k = 10$ provides a good trade-off.

This approximate solution highlights the conditions under which dPCA will work well, i.e., result in well-demixed components that capture most of the variance of the data: the main principal axes of different marginalizations $\mathbf{X}_{\phi}$ need to be non-collinear. In other words, principal subspaces of different marginalizations should not overlap.

## Comparison of dPCA with targeted dimensionality reduction

Next, we compare dPCA with 'targeted dimensionality reduction' (TDR), the method proposed by *Mante et al. (2013)*. Briefly, the algorithm underlying TDR works as follows:

1. Perform PCA of the trial-average neural data $\widetilde{\mathbf{X}}$ and define a 'denoising' matrix $\mathbf{K}$ as a linear projector on the space spanned by the leading principal axes. Here we used 20 principal axes: $\mathbf{K} = \mathbf{U}_{20}\mathbf{U}_{20}^{\top}$.
2. For each neuron $i$, regress its firing rate at each time point $t$ on stimulus, decision, and interaction between them:

$$x_i(t) = \beta_i^o(t) + \beta_i^s(t)s + \beta_i^d(t)d + \beta_i^{sd}(t)sd + \epsilon,$$

where $d$ is any suitable parametrization of decision, e.g. $d = \pm 1$, and $s$ is stimulus value (e.g. actual stimulation frequency in the somatosensory working memory task in monkeys).

3. Take $N$ values of $\beta_i^s(t)$ as defining an $N$-dimensional vector $\beta^s(t)$, and analogously define $\beta^d(t)$, $\beta^{sd}(t)$, and $\beta^o(t)$. *Mante et al. (2013)* did not use the condition-independent term $\beta^o(t)$ in their original study, but we treat it here on equal footing.

4. Project each $\beta^\phi(t)$ onto the space spanned by the leading PCA axes: $\mathbf{K}\beta^\phi(t)$.

5. For each of the three parameters select one vector $\beta_*^\phi = \mathbf{K}\beta^\phi(t_*^\phi)$, where $t_*^\phi$ is the time point at which the norm of $\mathbf{K}\beta^\phi(t)$ is maximal.

6. Finally, stack the obtained vectors to form a matrix $\mathbf{B} = [\beta_*^s \; \beta_*^d \; \beta_*^{sd} \; \beta_*^o]$ and perform QR decomposition $\mathbf{B} = \mathbf{QR}$ with $\mathbf{Q}$ orthonormal and $\mathbf{R}$ upper triangular to find a set of three orthogonal demixing axes (as columns of $\mathbf{Q}$). If the individual vectors in $\mathbf{B}$ are far from orthogonal, then the resulting axes will strongly depend on the order of stacking them into the matrix $\mathbf{B}$. We found that we obtain best results if we select this order *ad hoc* for each dataset; the orders we used were stimulus → decision → interaction for the somatosensory working memory task, stimulus → interaction → decision for the visuospatial working memory task, and interaction → decision → stimulus for both olfactory tasks. The vector $\beta^o(t)$ was always used last.

We applied TDR to all our datasets and observed that dPCA consistently outperforms it in terms of capturing variance and demixing task parameters. First, unlike dPCA, TDR yields only one component per task parameter. Second, even this component tends to retain more mixed selectivity than the corresponding dPCA component. Some representative components are shown in *Figure 14*.

## Comparison of dPCA with LDA

Linear Discriminant Analysis (LDA) is usually understood as a one-way technique: there is only one parameter (class id) associated with each data point, whereas in this manuscript we dealt with three parameters simultaneously. Therefore, LDA in its standard form cannot directly be applied to the demixing problem. We can, however, use the same data and covariance decomposition

$$\mathbf{X} = \sum_\phi \mathbf{X}_\phi + \mathbf{X}_{\text{noise}} \qquad\qquad \mathbf{C} = \sum_\phi \mathbf{C}_\phi + \mathbf{C}_{\text{noise}}$$

that dPCA is using and construct a separate LDA for each marginalization $\phi$. To the best of our knowledge, this framework does not have an established name, so we call it *factorial LDA*.

Let us first consider the case of finding demixed components for marginalization $\mathbf{X}_\phi$. We will denote the remaining part of the data matrix as $\mathbf{X}_{-\phi} = \mathbf{X} - \mathbf{X}_\phi$ and the remaining part of the covariance matrix as $\mathbf{C}_{-\phi} = \mathbf{C} - \mathbf{C}_\phi$. In turn, the goal of LDA will be to find linear projections that have high variance in $\mathbf{C}_\phi$ and low variance in $\mathbf{C}_{-\phi}$. In LDA, these matrices are usually called *between-class* and *within-class* covariance matrices (*Hastie et al., 2009*). The standard treatment of LDA is to maximize the multivariate signal-to-noise ratio

$$\text{tr}\Big(\mathbf{DC}_\phi \mathbf{D}^\top [\mathbf{DC}_{-\phi}\mathbf{D}^\top]^{-1}\Big),$$

where $\mathbf{D}$ is the matrix with discriminant axes in rows. The well-known solution is that $\mathbf{D}_{\text{LDA}}$ is given by the leading eigenvectors of $\mathbf{C}_{-\phi}^{-1}\mathbf{C}_\phi$ (stacked together as rows), or, equivalently, as eigenvectors of $\mathbf{C}^{-1}\mathbf{C}_\phi$.

More useful for our purposes is the reformulation of LDA as a reduced-rank regression problem (*Izenman, 2008*; *De la Torre, 2012*). When classes are balanced, it can be formulated as

$$L_{\text{LDA}} = \|\mathbf{G}_\phi - \mathbf{FDX}\|^2,$$

where $\mathbf{G}_\phi$ is a *class indicator matrix*. This matrix has as many rows as there are possible values of parameter $\phi$ and specifies which data point is labeled with which parameter value: $G_{ij} = 1$ if the $j$-th data point belongs to class $i$ (has $i$-th value of the parameter $\phi$) and $G_{ij} = 0$ otherwise. In the toy example shown in *Figure 2*, there are three classes with five points each, and so $\mathbf{G}_\phi$ will be a $3 \times 15$

matrix of zeros and ones. In this reformulation of LDA, the main interest is in the decoder matrix $\mathbf{D}$, whereas the encoder matrix $\mathbf{F}$, which serves to map the low-dimensional representation onto the class indicator matrix, plays only an auxiliary role.

In contrast, the dPCA loss function is

$$L_{\mathrm{dPCA}} = \|\mathbf{X}_\phi - \mathbf{FDX}\|^2,$$

where $\mathbf{X}_\phi$ is the matrix of the same size as $\mathbf{G}_\phi$ with $j$-th column being the class centroid of the class to which the $j$-th point belongs. This comparison highlights the difference between the two methods: LDA looks for decoders that allow to reconstruct class identity (as encoded by $\mathbf{G}_\phi$) whereas dPCA looks for decoders that allow to reconstruct class means (as encoded by $\mathbf{X}_\phi$). *Figure 2b,f,h* provides a toy example of a situation when these two goals yield very different solutions: the LDA projection separates the three classes better than the dPCA projection, but the dPCA projection preserves the information about the distance between classes.

Using the explicit solution for reduced-rank regression, one can show that $L_{\mathrm{LDA}}$ does indeed have eigenvectors of $\mathbf{C}^{-1}\mathbf{C}_\phi$ as a solution $\mathbf{D}_{\mathrm{LDA}}$ for decoder (see Section 8.5.3 in *Izenman, 2008*). Following the similar logic for $L_{\mathrm{dPCA}}$, one can derive the corresponding expression for the dPCA decoder: $\mathbf{D}_{\mathrm{dPCA}}$ is given by the eigenvectors of $\mathbf{C}^{-1}\mathbf{C}_\phi^2$ (personal communication with Maneesh Sahani).

A statistical test known as MANOVA can be seen as another possible factorial generalization of LDA. Given the same data and covariance decomposition, MANOVA tests if the effect of $\phi$ is statistically significant by analyzing eigenvalues of $\mathbf{C}_{\mathrm{noise}}^{-1}\mathbf{C}_\phi$. The eigenvectors of this matrix can in principle serve as decoders, but these projections are optimized to separate the contribution of $\phi$ from noise, not from the contributions of noise and other parameters. Hence, MANOVA is not the appropriate method for demixing purposes.

While the toy example of *Figure 2* illustrates that dPCA and LDA will in principle have very different solutions, we note that in all datasets considered here factorial LDA and dPCA yielded very similar components. This may reflect several peculiarities of the data: for instance, the population activity for different values of the same parameter was spaced rather evenly, and all decisions were binary. Nevertheless, we emphasize that dPCA is better suited for (demixed) dimensionality reduction due to its focus on reconstructing the original data, as explained and discussed in the Results (*Figure 2*).

## Comparison of dPCA with previous versions

Demixed PCA as presented here is conceptually based on our previous work. *Machens et al. (2010)* suggested a demixing method called *difference of covariances (DOC)* that can only handle two parameters, e.g. stimulus $s$ and time $t$. Given PSTHs $\mathbf{x}(s,t)$, DOC first constructs stimulus-dependent and time-dependent marginalizations $\bar{\mathbf{x}}(s) = \langle \mathbf{x}(s,t)\rangle_t$ and $\bar{\mathbf{x}}(t) = \langle \mathbf{x}(s,t)\rangle_s$, and then computes the difference between the stimulus and time covariance matrices $\mathbf{C}_s = \left\langle \bar{\mathbf{x}}(s)\bar{\mathbf{x}}(s)^\top \right\rangle$ and $\mathbf{C}_t = \left\langle \bar{\mathbf{x}}(t)\bar{\mathbf{x}}(t)^\top \right\rangle$,

$$\mathbf{S} = \mathbf{C}_s - \mathbf{C}_t.$$

Eigenvectors of S with maximum (positive) eigenvalues correspond to directions with maximum stimulus variance and minimum decision variance. Vice versa, eigenvectors with minimum (negative) eigenvalues correspond to directions with maximum decision variance and minimum stimulus variance. In the toy example presented in *Figure 2* DOC finds the axis that is very close to the first PCA axis of class centroids (which is also very close to the dPCA encoder axis shown on the figure), providing worse demixing than both LDA and dPCA.

A possible extension of DOC to more than two parameters is described in *Machens (2010)*. Here the PSTHs are assumed to depend on $M$ parameters, and the method constructs $M$ marginalizations by averaging over all parameters except one. The respective covariance matrices $\mathbf{C}_\phi$ are then formed as above. The extension of DOC seeks to find the matrix of orthogonal directions $\mathbf{U}$ such that

$$L = \sum_\phi \operatorname{tr}(\mathbf{U}_\phi^\top \mathbf{C}_\phi \mathbf{U}_\phi)$$

is maximized subject to $\mathbf{U}^\top\mathbf{U} = \mathbf{I}$ where $\mathbf{U} = [\mathbf{U}_1...\mathbf{U}_M]$. For $M = 2$ this can be shown to be equivalent to the original DOC. Note that *Machens (2010)* did not address the interaction terms.

The connection between the current dPCA and the DOC approach can be made more explicit if we consider the full covariance decomposition $\mathbf{C} = \sum_\phi \mathbf{C}_\phi$ and introduce into the dPCA loss function an additional constraint that both encoder and decoder should be given by the same matrix with orthonormal columns: $\mathbf{F}_\phi = \mathbf{D}_\phi^\top = \mathbf{U}_\phi$. Then

$$\begin{aligned}\|\mathbf{X}_\phi - \mathbf{U}_\phi\mathbf{U}_\phi^\top\mathbf{X}\|^2 &= \|\mathbf{X}_\phi - \mathbf{U}_\phi\mathbf{U}_\phi^\top\mathbf{X}_\phi\|^2 + \|\mathbf{U}_\phi\mathbf{U}_\phi^\top\mathbf{X}_{-\phi}\|^2\\ &= \|\mathbf{X}_\phi\|^2 - \|\mathbf{U}_\phi\mathbf{U}_\phi^\top\mathbf{X}_\phi\|^2 + \|\mathbf{U}_\phi\mathbf{U}_\phi^\top\mathbf{X}_{-\phi}\|^2\\ &\sim -\operatorname{tr}\big(\mathbf{U}_\phi^\top(\mathbf{C}_\phi - \mathbf{C}_{-\phi})\mathbf{U}_\phi\big),\end{aligned}$$

where the first equality follows from properties of the decomposition, the second equality from the properties of the orthonormal matrices $\mathbf{U}_\phi$, and the third equality uses the definition of the covariance. This derivation shows that the difference of covariances $\mathbf{C}_\phi - \mathbf{C}_{-\phi}$ emerges from the dPCA loss function if the decoder and encoder are given by the same set of orthogonal axes. However, such axes $\mathbf{U}_\phi$ from different marginalizations $\phi$ will in general not be orthogonal to each other, whereas both DOC and its generalization insisted on orthogonal axes.

Both the original DOC and its extension ignored interaction terms. *Brendel et al. (2011)* introduced interaction terms and the full covariance splitting $\mathbf{C} = \sum_\phi \mathbf{C}_\phi$ as described in this manuscript, and developed a probabilistic dPCA model based on probabilistic PCA (PPCA); to remove ambiguity we call this method dPCA-2011. Similar to PPCA, dPCA-2011 assumes that the data are described by a linear model with Gaussian residuals, i. e.

$$p(\mathbf{x}|\mathbf{z}) = \mathcal{N}(\mathbf{W}\mathbf{z}, \sigma^2\mathbf{I}),$$

but the prior over the components z is chosen such that the components are sparsely distributed over marginalizations. In other words, the prior is chosen such that those components are favored that have variance in only one marginalization. Under the constraint that decoding directions W are orthogonal, the model can be fit using the expectation-maximization algorithm. However, the probabilistic formulation of *Brendel et al. (2011)* still suffers from the orthogonality constraint. As explained in the Discussion, the orthogonality constraint is too rigid and can prevent successful demixing if parameter subspaces are sufficiently non-orthogonal. Indeed, we applied dPCA-2011 to all our datasets and observed that dPCA-2015 showed better demixing (*Figure 14*). Moreover, dPCA-2011 failed to find any decision components in the visuospatial working memory task.

In addition, the formulation of dPCA in this manuscript is radically simplified compared to *Brendel et al. (2011)*, features an analytic solution and is easier to compare with other linear dimensionality reduction techniques.

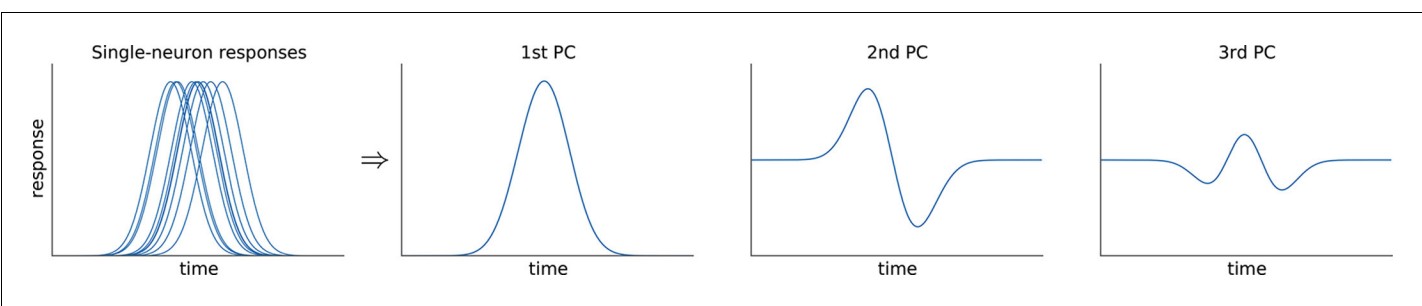

**Figure 15.** Fourier-like artifacts in PCA. (Left) In this toy example, single neuron responses are generated from the same underlying Gaussian but are randomly shifted in time. (Right) First three PCA components of the population data. While the leading component resembles the true signal, higher order components look like higher Fourier harmonics. They are artifacts of the jitter in time.

## Appendix A. Mathematical properties of the marginalization procedure

Above we presented marginalization procedure for three parameters. In order to generalize it for an arbitrary number of parameters, we introduce a more general notation. We denote as $\Psi$ the set of parameters (in the previous section $\Psi = \{t, s, d\}$; note that the trial index is not included into $\Psi$) and write $\bar{x}_\psi$ to denote a decomposition term that depends on a subset of parameters $\psi \subseteq \Psi$. In particular, $\bar{x}_\emptyset = \bar{x}$. In full analogy to the 3-parameter case, each term can be iteratively computed via

$$\bar{x}_\psi = \left\langle x - \sum_{\tau \subset \psi} \bar{x}_\tau \right\rangle_{\Psi \setminus \psi} = \langle x \rangle_{\Psi \setminus \psi} - \sum_{\tau \subset \psi} \bar{x}_\tau,$$

where $\langle \cdot \rangle_{\Psi \setminus \psi}$ denotes averaging over all parameters that are not elements of $\psi$ and averaging over the trial index. This equation can be rewritten in a non-iterative way by expanding the sum; this yields the expression with alternating signs that is similar to our ANOVA-style equations above:

$$\bar{x}_\psi = \sum_{\tau \subseteq \psi} (-1)^{|\tau|} \cdot \langle x \rangle_{(\Psi \setminus \psi) \cup \tau}. \qquad (\bigstar)$$

One can verify that this formula correctly describes the 3-parameter case presented above; the general case can be proven by induction. The noise term is defined via

$$x_{\text{noise}} = x - \sum_\psi \bar{x}_\psi = x - \langle x \rangle_\emptyset.$$

This decomposition has several useful properties. First, the average of any marginalization $\bar{x}_\psi$ over any parameter $\gamma \in \psi$ is zero. This can be seen from the equation $(\bigstar)$ because after averaging over $\gamma$ all terms will split into pairs with opposite signs (indeed, for each $\tau \ni \gamma$ there is another $\tau' = \tau \setminus \gamma$). Second, all marginalizations are pairwise uncorrelated, i.e. their covariance is zero: $\langle \bar{x}_\psi \bar{x}_\chi \rangle_\Psi = 0$. This can be seen from equation $(\bigstar)$ because $\bar{x}_\psi$ and $\bar{x}_\chi$ both consist of an even number of terms with alternating signs, so their product will also consist of an even number of terms with alternating signs, and after averaging over $\Psi$ all terms will become equal to $\pm \bar{x}^2$ and cancel each other. Third, from the definition of the noise term it follows that any marginalization $\bar{x}_\psi$ is uncorrelated with the noise term: $\langle \bar{x}_\psi x_{\text{noise}} \rangle_\Psi = 0$.

The fact that all marginalizations and the noise are pairwise uncorrelated allows to segregate the variance of $x$ (here we assume that $x$ is centered, i.e. $\bar{x} = 0$):

$$\text{var}[x] = \langle x^2 \rangle_\Psi = \left\langle \left( \sum_\psi \bar{x}_\psi + x_{\text{noise}} \right)^2 \right\rangle_\Psi = \sum_\psi \langle \bar{x}_\psi \rangle_\Psi + \langle x_{\text{noise}}^2 \rangle_\Psi = \sum_\psi \text{var}[\bar{x}_\psi] + \text{var}[x_{\text{noise}}].$$

Turning now to the multivariate case, if we replace $x$ with $\mathbf{x} \in \mathbb{R}^N$, everything remains true but variances should be replaced by covariance matrices:

$$\mathbf{C} = \langle \mathbf{x}\mathbf{x}^\top \rangle_\Psi = \sum_\psi \mathbf{C}_\psi + \mathbf{C}_{\text{noise}}.$$

Note that in ANOVA literature one usually talks about decomposing *sums of squares* $\sum x^2$ and in MANOVA literature about decomposing *scatter matrices* $\sum \mathbf{x}\mathbf{x}^\top$, because (co)variances of different terms are computed from these sums using different denominators (depending on the corresponding number of degrees of freedom) and do not add up. We do not make this distinction and prefer to talk about decomposing the (co)variance, i.e. all (co)variances here are defined with the same denominator equal to the total number of sample points.

## Appendix B. Fourier-like components from temporal variations

Consider the decision components in the somatosensory working memory task, *Figure 3*. Here the second and the third components are closely resembling the first and second temporal derivatives of the leading decision component. To illustrate why these components are likely to be artifacts of the underlying sampling process, consider a highly simplified example in which a population of $N$ neurons is encoding a one-dimensional bell-shaped signal $z(t)$ in the population vector $\mathbf{a}$, i.e the

population response is given by $\mathbf{y}(t) = \mathbf{a}z(t)$. In this case, the population response lies in the one-dimensional subspace spanned by $\mathbf{a}$ and the covariance matrix has rank one:

$$\mathbf{C} = \langle \mathbf{y}(t)\mathbf{y}^\top(t) \rangle_t = \mathbf{a}\mathbf{a}^\top \langle z(t)^2 \rangle_t.$$

Now consider the case in which the neurons are not recorded simultaneously but are pooled from different sessions. In behavioural experiments it is unavoidable that the onset of (self-timed) neural responses will vary by tenths or hundreds of milliseconds. Hence, the individual response $y_i(t)$ of neuron $i$ will experience a small time-shift $\tau_i$ so that $y_i(t) = a_i z(t + \tau_i)$, see *Figure 15* for an example with Gaussian tuning curves. If $\tau_i$ is small we can do a Taylor expansion around $t$,

$$y_i(t) = a_i z(t) + a_i \tau_i z'(t) + \mathcal{O}(\tau_i^2).$$

where we neglect higher-order corrections for simplicity, but the extension is straight-forward. Let $\tau$ be the vector of time-shifts of all neurons and let $\mathbf{b} = \mathbf{a} \circ \tau$ be the element-wise vector product of $\mathbf{a}$ and $\tau$, i.e. $[\mathbf{a} \circ \tau]_i = a_i \tau_i$. Then the population response can be written as

$$\mathbf{y}(t) \approx \mathbf{a}z(t) + \mathbf{b}z'(t).$$

Hence, the covariance matrix becomes approximately

$$\mathbf{C} \approx \mathbf{a}\mathbf{a}^\top \langle z^2(t) \rangle_t + \mathbf{b}\mathbf{b}^\top \langle z'^2(t) \rangle_t,$$

where we assumed for simplicity that $\mathbf{a} \perp \mathbf{b}$. In other words, time-shifts between observations will result in additional PCA components that roughly resemble the temporal derivatives of the source component.

## Data and code

The dPCA code is available at http://github.com/machenslab/dPCA for Matlab and Python. All four datasets used in this manuscript have been made available at http://crcns.org (*Romo et al., 2016*; *Constantinidis et al., 2016*; *Feierstein et al., 2016*; *Uchida et al., 2016*). Our preprocessing and the main analysis scripts (Matlab) are available at http://github.com/machenslab/elife2016dpca.

## Acknowledgements

This work was supported by the Bial Foundation, grant #389/14. We thank Matthew Kauffman, Valerio Mante, and Emmanuel Procyk for helpful input along the way, and Maneesh Sahani for a valuable discussion on the connection between dPCA and LDA. We furthermore thank Nuno Calaim for help with creating animations.

## Additional information

### Competing interests

NU: Reviewing editor, *eLife*. The other authors declare that no competing interests exist.

### Funding

| Funder | Grant reference number | Author |
| --- | --- | --- |
| Fundação Bial | Fellowship, 389/14 | Dmitry Kobak |

The funders had no role in study design, data collection and interpretation, or the decision to submit the work for publication.

### Author contributions

DK, WB, CKM, Conception and design, Analysis and interpretation of data, Drafting or revising the article; CC, CEF, AK, ZFM, X-LQ, RR, NU, Acquisition of data

## Author ORCIDs

Dmitry Kobak, http://orcid.org/0000-0002-5639-7209

Claudia E Feierstein, http://orcid.org/0000-0002-8002-922X

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
