## [Decision Letter]

Thank you for submitting your work entitled "Demixed principal component analysis of neural population data" for consideration by *eLife*. Your article has been reviewed by three peer reviewers, and the evaluation has been overseen by a Reviewing Editor and Eve Marder as the Senior Editor.

The reviewers have discussed the reviews with one another and the Reviewing editor has drafted this decision to help you prepare a revised submission.

Essential revisions:

As you will see from the individual reports below, there were rather divergent views on your manuscript. While we are not asking for additional analysis, we are asking for editorial revisions to make the method and its novelty more apparent.

We are asking for a major rewrite that 1) defines the method better 2) clarifies the difference and similarities with earlier dPCA approaches, 3) the interpretation of the results. Being a methods paper we feel that the presentation is particularly important. Also publication/public presentation of the code is a necessary requirement for *eLife* publication (this can be any of a number of public and maintained sites).

We are treating this as a Tools and Resources paper, as it is really a data analysis Methods paper, and should have been submitted under the TR heading.

Reviewer #1:

This paper is about an algorithm to analyze neural activity in response to experimental manipulations of various factors such as time, stimulus property, and behavioral response. There is a new technique proposed and discussed in the context of previous methods. The results of the different methods do not look all that different.

The main issue I have with the paper is that the method is described 3 times in 3 separate sections (Results, Methods, supplement), with increasing level of detail, but seemingly inconsistently in concept and notation. It is therefore very hard to make sense of it all.

At the end of the day it seems that the novel idea advocated here is rather simple: find a set of projections of the raw data which explain most of the variance in the mean response (mean over all trials and factors while holding one factor fixed). This in turn seems to be simply solved, for each factor separately, by an existing algorithm known as "reduced rank regression".

Complicating the description, the authors seem to have at least 3 algorithms which they call dPCA. First, one they proposed in 2011 under that exact title. Then, a second, apparently the new one, which is based on a cost function that was not entirely clear in the first reading in the Results section, but became clear only after reading the supplement. There is a third "approximate" version suggested, which concatenates the eigenvectors of the covariance for each mean response. (Is that what is called Naive demixing (Figure 12)?) They seem to argue at one point that this approximate method gives similar results to the superior new technique.

The methods seem to have all in common in their goal of explaining the covariance of the mean responses (mean across the factors, and at times, mean across trials) with a few dimensions. There is much talk about an additive model similar to what is used in conventional MANOVA, where the total data is explained as a sum of means across various combinations of factors. Though I don't see that this is necessary to motivate the dimensionality reduction proposed here, it does lead to comparison with yet one more technique. With the current presentation, I find it hard to keep it all straight.

In Figure 12 the various methods are then finally compared to each other, but as stated below, the performance metric presented there left me confused, just when I thought I knew what is going on. Importantly, the traces presented there don't seem all that different to each other. So then, what is the main new contribution of this paper?

Reviewer #2:

This manuscript gives a discursive presentation of how a new version of remixed principal component analysis (dPCA-2015) may be used to analyze multivariate neural data. The presentation is, I believe, complete enough that a user could reconstruct, from zero, the analysis method and the particulars of data pre-processing, etc. It is well-written and logical. The method itself is a nice compromise between a principal component approach to data analysis and a MANOVA-like approach. In particular, components are easily visualizable and the manuscript figures do a nice job of showing, at a glance, the results from fairly sophisticated experiments.

Other than a number of small textual changes that I would suggest, and one reference, I think this manuscript is in publishable form.

Reviewer #3:

This paper describes a statistical method, demixed-PCA, for finding low-dimensional "interpretable" structure in trial based neural recordings. In many cases, neurons exhibit "mixed selectivity", meaning that they exhibit tuning to multiple experimental variables (e.g., the visual stimulus and the upcoming decision). dPCA differs from PCA in that it tries to find a low-dimensional projection that separates ('demixes') the tuning to different kinds of variables. The paper applies dPCA to four different datasets from monkey PFC and rat OFC, and shows that it recovers structure consistent with previously reported findings using these datasets.

The paper is interesting and this technique is likely to be of considerable interest and importance to the field, particularly given recent interest in mixed selectivities in different brain areas. However, I have some concerns about novelty and about the intended contribution of this paper. A similar demixing approach has been described previously in Machens 2010 and Brendel 2011 (which was a NIPS paper, so perhaps shouldn't be counted against the novelty of this one since it's only a conference proceedings paper). But it would be helpful to describe a little bit more clearly the intended contribution. Is this just the journal-length writeup of the NIPS paper? How does the method described differ from that in the 2010 Frontiers paper?

It would also be nice to spell out a little bit more clearly what scientific insights the method is likely to provide (or indeed, provides for the datasets being analyzed here). It seemed that for each dataset, the paper mostly says that dPCA confirmed the findings in the original paper. But if that's the case, why do we need dPCA? What can we learn by analyzing data with this method that we didn't already have access to using the analysis methods from those papers?

I have two other high-level comments:

1) I think the authors don't do enough to describe how they decided which components to include when setting up the model. Presumably this is a choice made before running the method, i.e., how to map the elements of the experiment onto discrete conditions. How did the authors solve this problem, and how should potential users go about solving it when applying the method to new datasets. For example, why don't we have "reward" components for any of the datasets considered here? How did you decide which interaction terms to include in the first place? In the olfactory data, did you include different components for each mixture or just one for each mixing component? What are the keys to setting up the analysis? Are there any exploratory data analysis techniques that the authors used to make these decisions?

2) For a methods paper like this, providing code should be a mandatory requirement for publication. Please say something about where to find an implementation, so that would be users can try it out.

[Editors' note: further revisions were requested prior to acceptance, as described below.]

Thank you for resubmitting your work entitled "Demixed principal component analysis of neural population data" for further consideration at *eLife*. Your revised article has been favorably evaluated by Eve Marder (Senior editor), a Reviewing editor, and three reviewers. The manuscript has been greatly improved but there are some smaller remaining issues outlined below.

Reviewer #1 (General assessment and major comments (Required)):

This manuscript is like night and day compared to the previous version. I barely recognize the material any more. It is so much clearer that it makes me much more confident that this is all sound and well. I can hardly believe that these were the same authors.

Reviewer #1 (Minor Comments):

A few places where I got stuck were for example:

Equation 4 in subsection “Core dPCA: loss function and algorithm”, why can one separate the quadratic term like this? Also not clear why stefs 1-3 minimize this cost function, but I trust that all this is in the cited paper, though I could only retrieve the Izenman (1975) paper, which looked like yet more work to answer my questions.

Similarly, second equation in subsection “Unbalanced data”, why can one separate the square into two terms? and why can one replace X_noise_ by C^1/2^_noise_? and why is X_PSTH_=X tilde?

In general I have to say that, while the issue of balancing the data is intuitively clear, the corresponding math seems cumbersome, but again, this may be my lack of time.

Reviewer #2 (General assessment and major comments (Required)):

The revised manuscript is acceptable for publication.

Reviewer #3 (General assessment and major comments (Required)):

I thank the author for the detailed reply to the original review comments, and the revised manuscript is substantially improved. I have a few lingering technical comments and questions, but overall I think the paper is now suitable for publication.

Reviewer #3 (Minor Comments):

Figure 1: not entirely clear what principal components these are, i.e., why are there multiple traces per component? I guess this is because the authors have taken the principal components of the rows of the X matrix (i.e., each of which contains the PSTH of a single neuron for all components). This is a slightly odd choice: I would have thought one would take PCA of the PSTHs, which is closer in spirit to the dPCA and other methods discussed here, so that each component is just a length-T vector (i.e., where T is the number of time bins in the PSTH). I guess this is ok as is, but you should add something to the text or the caption to clarify what these are (e.g., each principal component is a PSTH across all conditions). This seems a little bit confusing, however, because the dPCA components you'll show later aren't the same size. (Or am I misinterpreting the figure, and what you're showing is instead the projection from all conditions on each PC?)

Indeed, the two components from Figure 1 explain only 23% of the total variance of the population firing rates and the two components from Figure 1 explain only 22% (see Methods). Consequently, a naive observer would not be able to infer from the components what the original neural activities looked like.

This seems a little bit uncharitable to the Mante et al. paper. After all, the regression model used by those authors did indeed have an "untuned component" for each neuron (referred to as Β-0), they simply didn't do anything with these components when it came time to construct a low-d projection that captured information about the stimulus and decision variables. But one could certainly have performed a dimensionality reduction of the Β_0's if one were interested in capturing information about time. So, in my view, while this passage is technically correct, I would encourage the authors to rephrase it to be slightly less disparaging to Mante et al. They simply weren't interested in the coding of time, so the fact that they don't capture those components is a choice about what variables to include more than a fundamental limitation of the method.

"where averaging takes place over all irrelevant parameters." A bit unclear – could use an extra sentence unpacking what this means.

"The overall variance explained by the dPCA components (Figure 3, red line) is very close to the overall variance explained by the PCA components (black line).": Why does PCA only get 80% of the variance? Is this because you've determined that the last 20% belongs to the noise component?

Figure 3 caption: "Thick black lines show time intervals during which the respective task parameters can be reliably extracted from single-trial activity". Does this mean a pseudotrial with 832 neurons? Start of this section mentions the # of neurons but I think it's important to say it here in the Figure caption what this means. (Presumably, if you have enough neurons recorded then single-trial decoding becomes perfect?)

---

## [Author Response]

Essential revisions:

As you will see from the individual reports below, there were rather divergent views on your manuscript. While we are not asking for additional analysis, we are asking for editorial revisions to make the method and its novelty more apparent.

We thank all the reviewers and the editors for their careful reading of our manuscript. The comments and suggestions were very helpful and have guided our efforts to clarify the manuscript and to improve the presentation. We provide a detailed rebuttal below.

Large portions of the text have been rewritten and rearranged, for that reason we found it difficult to provide a version of the manuscript with all changes tracked. The parts that are completely rewritten are (i) the presentation of dPCA method in the Results, (ii) the Methods. Please see the new manuscript file.

We are asking for a major rewrite that 1) defines the method better 2) clarifies the difference and similarities with earlier dPCA approaches, 3) the interpretation of the results. Being a methods paper we feel that the presentation is particularly important.

We rearranged, clarified and streamlined the presentation throughout the paper. To address point (1), we expanded and completely rewrote the presentation of the dPCA method in the Results section and also changed the accompanying figure. Furthermore, we merged the Supplementary Information with the Methods section, which now contains all the mathematical details. To address point (2), we now address these differences explicitly in both Introduction and Discussion. More importantly, we included a new section in the Methods that provides a detailed comparison between our older approaches and the current paper. To address point (3), we have clarified the description and layout of the figures taking into account all of the reviewers' comments. See below for more specific replies.

Also publication/public presentation of the code is a necessary requirement for eLife publication (this can be any of a number of public and maintained sites).

We fully support this policy. Indeed, we already published our code on github prior to the original submission. This was stated at the end of our Introduction.

“The dPCA code is available at http://github.com/wielandbrendel/dPCA for Matlab and Python.”

and we now re-iterate this information in the Methods section.

We are treating this as a Tools and Resources paper, as it is really a data analysis Methods paper, and should have been submitted under the TR heading.

We were not aware that *eLife* has a Tools and Resources section. We agree that this is where our manuscript belongs the best.

Reviewer #1:

This is a partial review only. I did not manage to understand all the pieces of this manuscript, and feel that without a few answers from the Authors, and a more streamlined presentation, I will struggle to provide a complete review. So I will list what I did and did not understand so far.

We thank the Reviewer for a thorough reading of our paper and for a lot of detailed comments. Based on the issues raised by the Reviewer, we have made a number of major changes in how we present the method and structure the text, see below.

This paper is about an algorithm to analyze neural activity in response to experimental manipulations of various factors such as time, stimulus property, and behavioral response. There is a new technique proposed and discussed in the context of previous methods. The results of the different methods do not look all that different.

We believe that this “lack of difference" is a misunderstanding that was caused by a lack of clarity from our side. Indeed, dPCA should mainly be compared with the methods presented in our Figure 1.e. with the methods that are actually being used in neuroscience to analyze and visualize neurophysiological recordings. The difference between dPCA and all these methods is obvious.

In contrast, the “previous methods" that the Reviewer mentions here probably refers to our original Figure 12, which compared our own preliminary approach dPCA-2011 (Brendel et al. 2011) to a variety of ad hoc demixing methods (such as MANOVA-based demixing", “naive demixing", etc.). However, none of these methods actually exist in the literature; rather, they were being presented here as alternative ways of doing something similar to dPCA with different, if sometimes subtle, trade-offs.

Our preliminary approach, dPCA-2011, was published as a conference proceedings paper, and is now substantially reformulated and improved (we extended our description in the Discussion of what are the differences and included a new section in the Methods); we do not claim that the difference between dPCA-2011 and dPCA-2015 is enormous, but we do argue that the improvement is quite noticeable and that the method, in its updated form and with an application to a variety of neural datasets, deserves a detailed journal-length treatment. Apart from changes in the method, we now provide an in-depth analysis of several experimental datasets, make a comparison between them, and clarify the relationship between dPCA and related methods such as LDA or the method of Mante et al. 2013. (Please see our largely updated presentation of LDA/dPCA in the Results and also the updated Discussion and Methods.)

All other approaches are ad hoc demixing methods; none of them has ever been used or even suggested for the analysis of neural data. We now see that what used to be our Figure 12 only caused confusion, so we removed all mentions of MANOVA-based demixing",”naive demixing", etc. as these methods do not exist in the literature. See below.

The main issue I have with the paper is that the method is described 3 times in 3 separate sections (Results, Methods, supplement), with increasing level of detail, but seemingly inconsistently in concept and notation. It is therefore very hard to make sense of it all.

The reviewer has a point. We undertook the following measures in order to address this issue:

1. We have merged the Supplementary Information with the Materials & Methods section.

2. We have largely expanded the presentation of dPCA in the Results, providing more geometrical intuitions, so that the main concepts are clearer.

3. We have reformatted our whole manuscript in LATEX which allowed us to use a consistent notation throughout the text.

The changes are too numerous to copy here entirely.

At the end of the day it seems that the novel idea advocated here is rather simple: find a set of projections of the raw data which explain most of the variance in the mean response (mean over all trials and factors while holding one factor fixed). This in turn seems to be simply solved, for each factor separately, by an existing algorithm known as "reduced rank regression".

That is correct. In fact, we are quite happy that the method has such a simple formulation and solution; we consider it a strength and not a weakness. Note that most of the related linear dimensionality reduction methods such as PCA, LDA, CCA, etc. can be formulated and solved just as easily.

Complicating the description, the authors seem to have at least 3 algorithms which they call dPCA. First, one they proposed in 2011 under that exact title. Then, a second, apparently the new one, which is based on a cost function that was not entirely clear in the first reading in the Results section, but became clear only after reading the supplement. There is a third "approximate" version suggested, which concatenates the eigenvectors of the covariance for each mean response. (Is that what is called Naive demixing (Figure 12)?) They seem to argue at one point that this approximate method gives similar results to the superior new technique.

It is true that the method published in 2011 in the NIPS conference proceedings has the same name; we decided that in the long run it will be less confusing if we keep using the name “dPCA" instead of inventing some other name. The core idea of the dPCA framework is to find linear projections of the full data that capture most of the variance and are demixed; dPCA-2011 and dPCA-2015 are two algorithmic approaches to solve this same task. DPCA-2015 supersedes the approach of 2011.

Note that this is not an uncommon situation; e.g. independent component analysis (ICA) is a name that encompasses a variety of related methods, and the same is true for sparse PCA or factor analysis (FA). In both cases there are various suggested implementations, but all of them are united by their goal.

We did not intend to advocate what we called “naive demixing" as a “third version". Rather, we intended to briefly describe this procedure in order to provide additional intuitions about dPCA and to explain a common misperception that we have encountered when talking to experimentalists. Since this is a minor technical point, we have downgraded the discussion of this “version" by eliminating it from the comparison figure, and by discussing it only briefly in a separate section in the methods (see section “Comparison of dPCA to PCA in each marginalization").

The methods seem to have all in common in their goal of explaining the covariance of the mean responses (mean across the factors, and at times, mean across trials) with a few dimensions. There is much talk about an additive model similar to what is used in conventional MANOVA, where the total data is explained as a sum of means across various combinations of factors. Though I don't see that this is necessary to motivate the dimensionality reduction proposed here, it does lead to comparison with yet one more technique. With the current presentation, I find it hard to keep it all straight.

We agree with the reviewer that the multitude of comparisons with other “methods", most of which do not even exist, was, with hindsight, more confusing than helpful. Essentially, all these comparisons were driven by conversations with e.g. someone claiming that what we are doing has been done by and is not different from e.g. MANOVA, so we felt we need to explain the difference. We have downgraded most of these discussions to just highlight the technical differences without giving each “ad hoc" method the prominence it had in the original submission. To straighten the reference to MANOVA, we have rewritten both the Discussion and Methods section about LDA/MANOVA.

In Figure 12 the various methods are then finally compared to each other, but as stated below, the performance metric presented there left me confused, just when I thought I knew what is going on. Importantly, the traces presented there don't seem all that different to each other. So then, what is the main new contribution of this paper?

We see the referee's point and we agree that Figure 12 was quite unfortunate. First, we reemphasize that two of the “methods" presented in Figure 12 were just ad hoc methods invented by us to contrast against dPCA (e.g. naive demixing or MANOVA-based demixing). Since this is indeed confusing, we eliminated these methods from the plot, leaving the only two published methods, dPCA- 2011 and targeted dimensionality reduction by Mante et al., 2013 (Figure 14 in the revised version). Second, we notice that there are differences between the remaining methods, even though they may appear subtle. dPCA-2011 fails to find one component; the method of Mante et al. does not produce any condition-independent components. The differences are not enormous, but noticeable and consistent.

It's especially strong where the encoding subspaces are not orthogonal. Third, we note that applying these methods to data is also not necessarily the best way of showing how they differ – too much depends on the specifics of the data set. To address these problems, we completely rewrote our explanation of dPCA and added additional geometric intutions that show how dPCA differs from PCA and LDA. We hope that these changes clarify the contributions of dPCA-2015: a method that is far simpler and more direct than any of the existing (or ad hoc) methods, and that achieves overall better demixing, although, as the reviewer noticed, the differences on real data are not necessarily huge.

Concerning the performance metric, we compare all methods using linear projections of the full data and the variance is the variance of this projection.

Reviewer #2:

I am not an experimentalist, so this review will have nothing to say about the details of the experimental design, etc. I was not able to view the video(s) with my QuickTime viewer, so I also have nothing to say about the video(s).

This manuscript gives a discursive presentation of how a new version of remixed principal component analysis (dPCA-2015) may be used to analyze multivariate neural data. The presentation is, I believe, complete enough that a user could reconstruct, from zero, the analysis method and the particulars of data pre-processing, etc. It is well-written and logical. The method itself is a nice compromise between a principal component approach to data analysis and a MANOVA-like approach. In particular, components are easily visualizable and the manuscript figures do a nice job of showing, at a glance, the results from fairly sophisticated experiments.

Other than a number of small textual changes that I would suggest, and one reference, I think this manuscript is in publishable form.

We thank the reviewer for these comments.

Reviewer #3:

This paper describes a statistical method, demixed-PCA, for finding low-dimensional "interpretable" structure in trial based neural recordings. In many cases, neurons exhibit "mixed selectivity", meaning that they exhibit tuning to multiple experimental variables (e.g., the visual stimulus and the upcoming decision). dPCA differs from PCA in that it tries to find a low-dimensional projection that separates ('demixes') the tuning to different kinds of variables. The paper applies dPCA to four different datasets from monkey PFC and rat OFC, and shows that it recovers structure consistent with previously reported findings using these datasets.

The paper is interesting and this technique is likely to be of considerable interest and importance to the field, particularly given recent interest in mixed selectivities in different brain areas.

We thank the Reviewer for these comments.

However, I have some concerns about novelty and about the intended contribution of this paper. A similar demixing approach has been described previously in Machens 2010 and Brendel 2011 (which was a NIPS paper, so perhaps shouldn't be counted against the novelty of this one since it's only a conference proceedings paper). But it would be helpful to describe a little bit more clearly what the intended contribution. Is this just the journal-length writeup of the NIPS paper? How does the method described differ from that in the 2010 Frontiers paper?

We understand that this history of publications makes it look as if we are trying to publish the same method over and over again. That is not the case. Rather, we see the current paper as the culmination of our efforts in developing demixing dimensionality reduction. It is similar in spirit to the previous papers, but technically very different. Briefly, here is our reasoning: Machens et al. 2010 (JNeurosci) introduced a “difference of covariances" method that can work with only two parameters, and has therefore rather limited applicability. Machens 2010 (Frontiers) papers offered one way to generalize that method to more parameters, but only in some specific cases; and it did not properly capture interaction terms. Indeed, this latter problem made its application to real data rather unattractive (unpublished observations). Finally, the Brendel et al. 2011 NIPS paper suggested a fully general way of splitting the data matrix and the covariance matrix into additive parts and then offered a probabilistic model to find the demixing projections. The latter paper also introduced the name “demixed principal component analysis".

The NIPS paper was a purely technical paper, and, back then, for us, the end of method development. At this point, we knew that dPCA worked very nicely on the Romo data, which is only briefly discussed within the NIPS paper. However, for a journal-length writeup, we intended to go beyond the Romo data to demonstrate the applicability of dPCA-2011 on a wider range of data sets. Alas, when we applied dPCA-2011 to other datasets, we saw that the demixing was not as good as in the Romo data. Further investigation of this problem showed that the orthogonality constraint of dPCA-2011, i.e., our emphasis on finding demixed components that are orthogonal to each other, posed a problem in some of the cases. Hence, we needed a more flexible approach, which caused us to revisit the method and design dPCA-2015.

This history is probably not very interesting for the general reader. However, we agree with the reviewer that it is important to explain to readers how the methods differ. We have now included a more explicit paragraph into the Introduction, extended a corresponding subsection in the Discussion, and wrote a new section in the Methods that explicitly contrasts the older methods with dPCA-2015.

It would also be nice to spell out a little bit more clearly what scientific insights the method is likely to provide (or indeed, provides for the datasets being analyzed here). It seemed that for each dataset, the paper mostly says that dPCA confirmed the findings in the original paper. But if that's the case, why do we need dPCA? What can we learn by analyzing data with this method that we didn't already have access to using the analysis methods from those papers?

The key problem that someone who has recorded 100s^-1^000s of neurons in a behavioral task faces is how to make sense of the incredible diversity of responses. The classical solutions are summarized in our Figure 1, and they essentially focus on extracting features of the data. Through clever combination of these classical techniques, researchers have always been able to extract important, or even key features in their data. The reviewer is correct in noting that dPCA is not guaranteed to discover anything beyond that, and, in the data sets we analyzed, we mostly confirmed the findings of the original studies (see below for some differences). So why dPCA?

There are several answers to this question. First, the classical techniques are actually quite laborious, and they require a lot of intuitions and work on the side of the researchers. Which part of the data to focus on first? Which time-periods or which parameters? dPCA is supposed to radically simplify this process by allowing researchers to boil down complex data sets and represent them in a few components, or, in essence, print out pretty much everything that's happening on a single page. First and foremost, dPCA is therefore a quick visualization method for the data, something that can serve as an entry point to analyzing certain aspects of the data in more depth and detail.

Second, dPCA is designed to not `lose' any important part of the data, where importance is measured in terms of neural variability, i.e., changes in _ring rates. Our emphasis on (approximately) loss-less dimensionality reduction or compression highlights several features of the data that have mostly been disregarded in the past (although they may be somewhat familiar at least to the people who have access to the raw data): these include the large percentage of neural variance falling into the condition independent components, the lack of separate cell classes, the precise dynamics of the embedding of the task parameters in the neural population response etc, see the section “Universal features of the PFC population activity". Of course, the question then is, why should we care about these features? That brings us to the next point:

Third, dPCA is designed to facilitate the comparison of population activity across data sets, which is one of the main emphasis of this paper. Since there is no unique way of applying the classical methods (Figure 1; statistical significance, regression, etc.) to neural population data, different studies usually rely on different combination of methods. In turn, it becomes essentially hard to impossible for a reader to appreciate the similarities and/or differences of monkey PFC data in different conditions, especially if the studies were carried out in different labs. dPCA provides a simple, standardized way of visualizing the data, which in turn does allow one to compare neural activities across data sets. In turn, if certain dominating aspects re-occur, such as the condition-independent components, or the dynamics of the representation of information, then maybe these aspects deserve special attention, because they tell us something about what these areas are doing.

I have two other high-level comments:

1) I think the authors don't do enough to describe how they decided which components to include when setting up the model. Presumably this is a choice made before running the method, i.e., how to map the elements of the experiment onto discrete conditions. How did the authors solve this problem, and how should potential users go about solving it when applying the method to new datasets. For example, why don't we have "reward" components for any of the datasets considered here? How did you decide which interaction terms to include in the first place? In the olfactory data, did you include different components for each mixture or just one for each mixing component? What are the keys to setting up the analysis? Are there any exploratory data analysis techniques that the authors used to make these decisions?

This is a very important issue, especially with respect to the goals of dPCA as explained above, and we updated both our Discussion and the methods (sections on balanced and unbalanced data and missing data) to cover it in more depth. To answer the questions above:

Why are there no “reward" components? In the monkey datasets we analyzed only correct trials because we lacked data on error trials since the monkeys often did not make enough mistakes. Accordingly, reward category is not very meaningful. In the olfactory datasets, we analyzed both correct and incorrect trials. Here, however, the presence/absence of reward is fully determined by stimulus and decision, so the “reward" label is superfluous given stimulus and decision labels. We note, however, that the activity related to reward shows up in the “interaction" terms, precisely because reward is based on the interaction of stimulus and decision.

How did you decide which interaction terms to include? Since we had stimulus and decision labels in all datasets, there really is only one interaction term. This interaction term was always included. (Time plays a special role in neurophysiological data and is treated differently as a parameter, as explained in the dPCA section in the Results and Methods.)

In the olfactory data, did you include different components for each mixture or just one for each mixing component? In the Kepecs et al. olfactory dataset, different odour mixtures were simply treated as different stimuli.

What are the keys to setting up the analysis? Are there any exploratory data analysis techniques that the authors used to make these decisions? The question of what labels to include is indeed very meaningful. We can provide some guidelines, but not a precise step-by-step recipe, as this depends too much on the scientific question addressed by an experiment. For instance, one could e.g. have some extra label available (imagine that for each stimulus and decision there were several possible values of reaction time, or movement type, etc.). Should they be included in the dPCA analysis as another parameter? That depends on the question, and there are some tradeoffs involved. As the number of task parameters included increases, so does the number of parameters (decoders and encoders) in dPCA, especially since there will be more and more interaction terms. Hence, the statistical power decreases and so does the usefulness of dPCA. Here is our updated Discussion:

“Second, the number of neurons needs to be sufficiently high in order to obtain reliable estimates of the demixed components. In our datasets, we found that at least _100 neurons were needed to achieve satisfactory demixing. The number is likely to be higher if more than three task parameters are to be demixed, as the number of interaction terms grows exponentially with the number of parameters. This trade-off between model complexity and demixing feasibility should be kept in mind when deciding how many parameters to put into the dPCA procedure. In cases when there are many task parameters of interest, dPCA is likely to be less useful than the more standard parametric single-unit approaches (such as linear regression). As a trivial example, imagine that only N = 1 neuron has been recorded; it might have strong and significant tuning to various parameters of interest, but there is no way to demix (or decode) these parameters from the recorded population."

2) For a methods paper like this, providing code should be a mandatory requirement for publication. Please say something about where to find an implementation, so that would be users can try it out.

We published our code on github prior to the original submission, which is stated in the end of our Introduction. We now included a reminder of this in the Methods.

“The dPCA code is available at http://github.com/wielandbrendel/dPCA for Matlab and Python.”

[Editors' note: further revisions were requested prior to acceptance, as described below.]

The manuscript has been greatly improved but there are some smaller remaining issues outlined below.

Reviewer #1 (General assessment and major comments (Required)):

This manuscript is like night and day compared to the previous version. I barely recognize the material any more. It is so much clearer that it makes me much more confident that this is all sound and well. I can hardly believe that these were the same authors.

We are very happy to hear that and thank the reviewer for the valuable comments.

Reviewer #1 (Minor Comments):

A few places where I got stuck were for example:

Equation 4 in subsection “Core dPCA: loss function and algorithm”, why can one separate the quadratic term like this? Also not clear why stefs 1-3 minimize this cost function, but I trust that all this is in the cited paper, though I could only retrieve the Izenman 1975 paper, which looked like yet more work to answer my questions.

We updated this section to make it as self-contained as possible and to avoid sending readers to the original papers from the 1970s. Reduced-rank regression is quite obscure in our field and we have therefore tried to clarify all the steps as much as possible. Please let us know if something remains unclear here.

The updated paragraphs now read:

“We note that the loss function *L_Ø_* is of the general form ||**X**_Ø_ – **AX**||^[2]^, with **A** = **FD**. For an arbitrary *N x N* matrix **A**, minimization of the loss function amounts to a classical regression problem with the well-known ordinary least squares (OLS) solution, A_OLS_ =**X**_Ø_**X**^T^ (**XX**^T)-1^. In our case, **A** = **FD** is an *N x N* matrix of rank *q*, which we will make explicit by writing **A**_q_. The dPCA loss function therefore amounts to a linear regression problem with an additional rank constraint on the matrix of regression coefficients. This problem is known as reduced-rank regression (*RRR*) (Izenman, 1975; Reinsel and Velu, 1998; Izenman, 2008) and can be solved via the singular value decomposition.

To see this, we write **X**_Ø_ –**A**_q_**X** = (**X**_Ø_ –**A**_OLS_**X**) + (**A**_OLS_**X** – **A**_q_**X**). The first term, **X**_Ø_ –**A**_OLS_**X**, consists of the regression residuals that cannot be accounted for by any linear transformation of X. It is straightforward to verify that these regression residuals, **X**_Ø_ –**A**_OLS_**X**, are orthogonal to **X** (Hastie et al., 2009, Section 3.2) and hence also orthogonal to (**A**_OLS_ – **A**_q_)X. This orthogonality allows us to split the loss function into two terms,

|| **X**_Ø_ –**A**_q_**X** ||^[2]^ = || **X**_Ø_ –**A**_OLS_**X**||**^[2]^**+|| **A**_OLS_**X – A**_q_**X**||^[2]^

where the first term captures the (unavoidable) error of the least squares fit while the second term describes the additional loss suffered through the rank constraint. Since the first term does not depend on Aq, the problem reduces to minimizing the second term.

To minimize the second term, we note that the best rank-q approximation to **A**_OLS_**X**

is given by its first *q* principal components (Eckart-Young-Mirsky theorem). Accordingly, if we write ***U****_q_* for the matrix of the *q* leading principal directions (left singular vectors) *u_i_* of **A**_OLS_**X**, then the best approximation is given by **U**_q_**U**^T^_q_**A**_OLS_**X** and hence **A**_q_ =**U**_q_**U**^T^_q_**A**_OLS_.

To summarize, the reduced-rank regression problem posed above can be solved in a three-step procedure: […]”

Similarly, second equation in subsection “Unbalanced data, why can one separate the square into two terms? and why can one replace X_noise_ by C^1/2^_noise_? and why is X_PSTH_=X?

We have expanded the explanation of this bit as follows:

“In the balanced case, the dPCA loss function *L_Ø_* can be rewritten as the sum of two terms with one term depending on the PSTHs and another term depending on the trial-to-trial variations,

*L_Ø_* = ||**X***_Ø_* – **FDX**||^[2]^ = ||**X***_Ø_
*– **FD (X** – **X**_noise_)||^[2]^ + ||**FDX**_noise_||^[2]^;

where we used the fact that **X***_Ø_* and **X** – **X**_noise_ are orthogonal to **X**_noise_ (see Appendix A). We now define **X**_PSTH_ = X – X_noise_ which is simply a matrix of the same size as **X** with the activity of each trial replaced by the corresponding PSTH. In addition, we observe that the squared norm of any centered data matrix **Y** with *n* data points can be written in terms of its covariance matrix **C**_Y_ = **YY**^T^/*n*, namely ||**Y**||^[2]^ = tr[**YY**^T^] = *n* tr[**C**_Y_] = *n* tr[**C**^1/2^
_Y_**C**^1/2^
_Y_] = *n*|| **C**^1/2^
_Y_ ||^[2]^, and so

*L_Ø_* = ||**X***_Ø_* – **FDX**_PSTH_||^[2]^ + *K SQT*|| **FCD**^½^_noise_||^[2]^

The first term consists of *K* replicated copies: **X**_PSTH_ contains *K* replicated copies of X (which we defined above as the matrix of PSTHs) and X*_Ø_* contains *K* replicated copies of X*_Ø_
*(which we take to be a marginalization of **X**, with **X** = Ʃ*_Ø_
*X*_Ø_*). We can eliminate the replications and drop the factor *K* to obtain

L*_Ø_* = ||X*_Ø_
*– **FDX** ||^[2]^ + *SQT* ||**FDC^[1]^**^/2^_noise_||^[2]^”

In general I have to say that, while the issue of balancing the data is intuitively clear, the corresponding math seems cumbersome, but again, this may be my lack of time.

The balancing math can indeed appear somewhat confusing, however we should point out that this whole part is not essential for the understanding of our method. We included the full treatment of the balancing issue because it can be important in practical applications when the conditions are very unbalanced. Otherwise, we can expect the direct formulation of dPCA without any rebalancing modifications with ||**X***
_Ø_
*–**FDX**||^[2]^ loss function to work fine, and the generalization to the loss function with PSTH matrices ||X*
_Ø_* – **FDX**||^[2]^ for the sequentially recorded datasets is intuitive.

*Reviewer #2 (General assessment and major comments (Required)): The revised manuscript is acceptable for publication. Reviewer #2 (Minor Comments): No author was designated with author association superscript 6 – Harvard University.*

Author designation with superscript 2 is not provided.

- There's probably an error in the numbering of author associations.

We fixed the affiliation list.

Reviewer #3 (General assessment and major comments (Required)):

I thank the author for the detailed reply to the original review comments, and the revised manuscript is substantially improved. I have a few lingering technical comments and questions, but overall I think the paper is now suitable for publication.

Reviewer #3 (Minor Comments):

Figure 1: not entirely clear what principal components these are, i.e., why are there multiple traces per component? I guess this is because the authors have taken the principal components of the rows of the X matrix (i.e., each of which contains the PSTH of a single neuron for all components). This is a slightly odd choice: I would have thought one would take PCA of the PSTHs, which is closer in spirit to the dPCA and other methods discussed here, so that each component is just a length-T vector (i.e., where T is the number of time bins in the PSTH). I guess this is ok as is, but you should add something to the text or the caption to clarify what these are (e.g., each principal component is a PSTH across all conditions). This seems a little bit confusing, however, because the dPCA components you'll show later aren't the same size. (Or am I misinterpreting the figure, and what you're showing is instead the projection from all conditions on each PC?)

There is some confusion here. As it might partially be due to the terminology (“principal axes" vs “principal components"), we should clarify it here. The PSTH matrix **X** has *N* rows (*N* is the number of neurons) and *SQT* columns (number of data points in the PSTHs in all conditions, *S* is the number of stimuli, *D* the number of decisions, *T* the number of time-points). We consider this as *SQT* points in the *N*-dimensional space and perform PCA on these data. Covariance matrix is *N x N*, each eigenvector (a direction in the *N*-dimensional space) defines a principal axis, and the projection onto this axis we call “principal component" and it has *SQT* points. These projections are shown on Figure 1. We clarified this in the legend and also in the Methods section “Implementation of classical approaches".

(Perhaps by “PCA of the PSTHs" the reviewer means reshaping these data as *N SQ* different PSTHs of length *T*? Then each PC would indeed be of length *T*; but we want to work in the *N*-dimensional (and not *NSQ*-dimensional) space of neurons because we want PCA/dPCA projections to be interpreted as linear readouts from the neural population.)

We furthermore emphasize that in the dPCA treatment we always work in the same *N*-dimensional space. In all subsequent figures (e.g. Figure 3), we always display all conditions in each panel, so we emphasize that the size of the PCA/dPCA components does not change throughout the manuscript. Indeed, although the condition-independent component may sometimes appear to consist of only one line, for instance, we are plotting one line for each condition, and they are just on top of each other (as they should be when the components are properly demixed).

*Indeed, the two components from Figure 1 explain only 23% of the total variance of the population firing rates and the two components from Figure 1 explain only 22% (see Methods). Consequently, a naive observer would not be able to infer from the components what the original neural activities looked like. This seems a little bit uncharitable to the Mante* et al. *paper. After all, the regression model used by those authors did indeed have an "untuned component" for each neuron (referred to as Β-0), they simply didn't do anything with these components when it came time to construct a low-d projection that captured information about the stimulus and decision variables. But one could certainly have performed a dimensionality reduction of the Β_0's if one were interested in capturing information about time. So, in my view, while this passage is technically correct, I would encourage the authors to rephrase it to be slightly less disparaging to Mante* et al.

*They simply weren't interested in the coding of time, so the fact that they don't capture those components is a choice about what variables to include more than a fundamental limitation of the method.*

This is a fair point. Strictly speaking, Mante et al. define their targeted dimensionality reduction (TDR) procedure (in the supplementary materials to their paper) by using only linear terms in the regression, and they disregard the *β*0-term. However, the reviewer is correct that one could treat _0 in the same way as other regression terms, and we thank the reviewer for pointing this out. More generally, as with any supervised or regression-based method, there are many possible extensions of TDR one could envision (e.g. including quadratic terms into the regression, using several time-points instead of choosing only one time-point, etc.). However, our point here was to contrast the trade-o_ between supervised and unsupervised methods, and so we chose to present TDR exactly as it was originally introduced.

To address the reviewer's comments, we have now updated our “Comparison of dPCA with targeted dimensionality reduction" section in the Methods to be clear about the possibility of using *β*0, and updated our Figure 14 to show these condition-independent components produced by TDR. We do not think the introductory Results section is the right place to elaborate on possible extensions of TDR, but we have slightly reworded the section to be more careful about our comparison of the supervised and unsupervised methods, and we now write:

“While such supervised approaches can be extended in various ways to produce more components and capture more variance, a more direct way to avoid this loss of information is to resort to unsupervised methods such as principal component analysis (PCA).”

To emphasize the importance of having more than one component per task parameter, we have now prepared another Video and included one additional paragraph in the Results section describing our first dataset:

“The first stimulus component (#5) looks similar to the stimulus components that we obtained with standard regression-based methods (Figure 1) but now we have further components as well. Together they show how stimulus representation evolves in time. In particular, plotting the first two stimulus components against each other (see Video 1) illustrates how stimulus representation rotates in the neural space during the delay period so that the encoding subspaces at F1 and F2 periods are not the same (but far from orthogonal either).”

"where averaging takes place over all irrelevant parameters." A bit unclear – could use an extra sentence unpacking what this means.

We slightly changed the formulation of this bit; what is meant here is illustrated in the next sentence using the toy example:

“First, we require that the compression and decompression steps reconstruct not the neural activity directly, but the neural activity averaged over trials and over some of the task parameters. In the toy example, the reconstruction target is the matrix of stimulus averages, **X**s, which has the same size as **X**, but in which every data point is replaced by the average neural activity for the corresponding stimulus, as shown in Figure 2.”

This averaging is also explained more formally below and is illustrated on Figure 2.

*"The overall variance explained by the dPCA components (Figure 3, red line) is very close to the overall variance explained by the PCA components (black line)." Why does PCA only get 80% of the variance? Is this because you've determined that the last 20% belongs to the noise component?*

The location of the legend was suboptimal on this figure panel and made it look as if “PCA" label refers to the horizontal dashed line at 80%. Instead, “PCA" label refers to the black line of cumulative explained variance that can be hardly seen because of the red dPCA line. So the correct interpretation is that PCA explained variance grows with the number of components and reaches 80% at 15 components. With more components it would of course exceed 80%. At the same time, the dashed horizontal line shows our estimate of the amount of “signal variance", i.e. we estimate that 20% of the variance is due to the noise in our PSTH estimates. This computation is explained in the Methods, but is not essential for anything else.

We have changed the legend location to remove the ambiguity.

*Figure 3 caption: "Thick black lines show time intervals during which the respective task parameters can be reliably extracted from single-trial activity". Does this mean a pseudotrial with 832 neurons? Start of this section mentions the # of neurons but I think it's important to say it here in the Figure caption what this means. (Presumably, if you have enough neurons recorded then single-trial decoding becomes perfect?)*

Yes, the decoding was done using pseudotrials with 832 neurons. The exact number does not seem to be important in the _gure caption: we simply used all available neurons. But the reviewer is right in that it is important to point out that the decoding was done using pseudotrials (and hence ignores noise correlations). We clarified the figure caption as follows:

“Thick black lines show time intervals during which the respective task parameters can be reliably extracted from single-trial activity (using pseudotrials with all recorded neurons)”